# Factual and Personalized Recommendations using Language Models and Reinforcement Learning

## Abstract

Recommender systems (RSs) play a central role in connecting users to content, products, and services, matching candidate items to users based on their preferences. While traditional RSs rely on implicit user feedback signals, conversational RSs interact with users in natural language. In this work, we develop a *comPelling*, *Precise*, *Personalized*, *Preference-relevant* language model (P⁴LM) that recommends items to users while putting emphasis on explaining item characteristics and their relevance. P⁴LM uses the *embedding space* representation of a user's preferences to generate compelling responses that are factually-grounded and relevant w.r.t. the user's preferences. Moreover, we develop a joint reward function that measures precision, appeal, and personalization, which we use as AI-based feedback in a reinforcement learning-based language model framework. Using the MovieLens 25M dataset, we demonstrate that P⁴LM delivers compelling, personalized movie narratives to users.

## 1 Introduction

Recommender systems (RSs) have emerged as a dominant way in which users discover content, products, and services (Resnick & Varian, 1997). Traditional RSs match candidate items to users based on their estimates for items preferences, possibly conditioned on some query or context. However, these preferences are often based on implicit user behavioral signals, such as clicks, number of watches, ratings, purchases, etc. Unfortunately, these provide little opportunity for an RS to elicit high-bandwidth preference information from users, explain recommendations, or for users to critique and steer their interaction with the RS. *Conversational RSs* have therefore attracted considerable attention as means to use natural-language interaction to facilitate more effective communication between RSs and their users (Sun & Zhang, 2018; Lei et al., 2020; Shen et al., 2023).

The emergence of language models (LMs) as a powerful paradigm for user engagement (Li et al., 2018; Friedman et al., 2023) suggests their use as a vehicle for conversational RSs. However, this requires LMs to engage in a personalized manner, adhering to users' preferences. In this paper, we explore the intersection of RSs and LMs, and more particularly, the use of LMs to enrich the user experience in RSs. We develop techniques which allow an LM to communicate the nuances of recommended items to a user, detailing their features, benefits, and explaining their *alignment with a user's preferences*. Such *personalized LMs* are not meant to "convince" users in the traditional sense, but rather, to articulate the *genuine and relevant merits* of a recommended item relative to the user.

Personalized LMs offer users a fully tailored RS experience, ensuring they find what they truly need and value. However, a number of challenges must be addressed in this endeavor: (i) any recommended item should be predicted to have maximal value given the user's preferences; (ii) the integrity and accuracy of an item's information is paramount; (iii) the personalized LM should present a reasonably comprehensive portrayal of the item by describing its merits and drawbacks, with a focus on *relevance* to the user's preferences; (iv) and finally, the LM's explanations or endorsements should be compelling and appealing to the user, provided that it meets the other criteria. In this work, we develop a framework centered around these four principles.

A key question we addressed in this work is how to effectively utilize the information captured by an RS embedding space to generate a factual, personalized, compelling, and relevant recommendations.

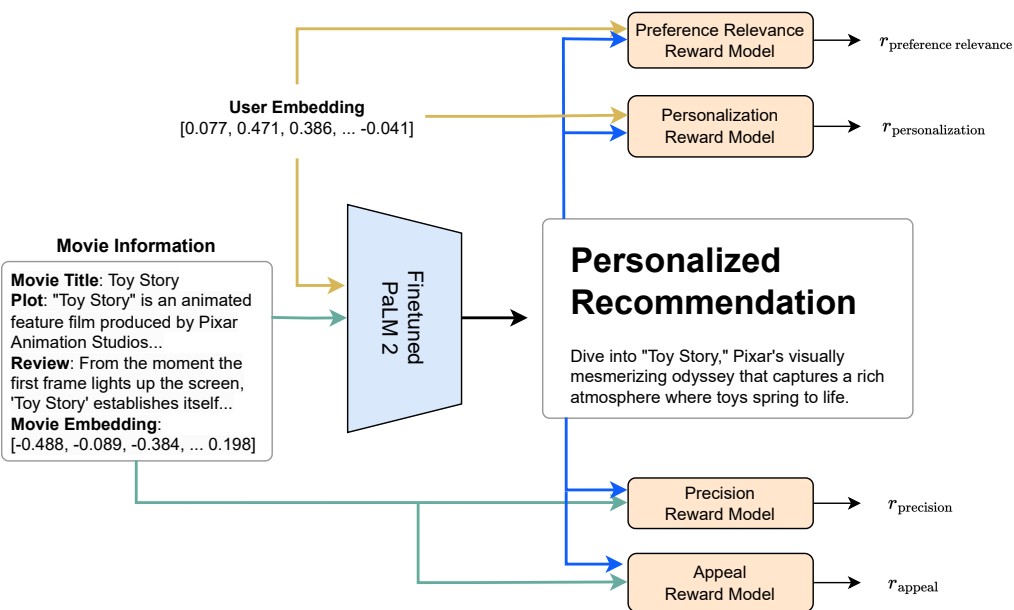

Figure 1: The P⁴LM Learning Framework for Recommendation Endorsement Generations.

Our contributions are three-fold. First, we quantify the aforementioned four attributes using reward functions, enabling systematic evaluation. Second, leveraging recent advances in reinforcement learning from AI feedback (RLAIF) (Lee et al., 2023), we develop an LM fine-tuning methodology to better align with these four rewards (see Figure 1 for the schematic diagram illustrating the RLAIF framework). Our developed model, which we term P⁴LM, not only comprises semantic skills, but also understands users' preferences encoded in the RS embedding space, providing factual, compelling, personalized endorsements. Finally, building on the MovieLens 25M dataset (Harper & Konstan, 2015) we showcase the potential of P⁴LM , powering a conversational movie recommender that promotes customized, relevant, and holistic interactions for users.

We begin with a brief introduction of RSs, LMs and the use of contextual Markov decision processes (CoMDPs) for modeling generative language problems of RSs (Section 2). We then describe the four principles, (i.e., personalization, precision, appeal, and preference relevance), which we incorporate into training of LMs for RSs (Section 3), followed by an reinforcement learning based fine-tuning methodology for training P⁴LM (Sections 4). Finally, we demonstrate the effectiveness of P⁴LM in generating factual, personalized, and compelling movie endorsement narratives for users within the MovieLens 25M benchmark dataset (Section 5).

## 2    PRELIMINARIES

In this section we present some basic background, outline our problem formulation, and establish the terminology used throughout the paper.

**Recommender Systems (RSs).**    To model user-item behavioral relationships in a personalized RS, we assume a standard collaborative filtering (CF) task (Su & Khoshgoftaar, 2009). Collaborative filtering finds similar patterns among users, filtering out items based on ratings of similar users. Given a user $u \in \mathcal{U}$, we use $r_{u,i}$ (e.g., 1–5 stars) to denote the rating of item $i \in \mathcal{I}$ by user $u$. Let $\mathcal{R}$ denote the $|\mathcal{I}| \times |\mathcal{U}|$ (usually sparse) ratings matrix corresponding to the *ratings dataset* $\mathcal{R} = \{(u, i, r_{u,i}) : r_{u,i} \neq 0\}$. To predict users' preference behavior, an RS learns user and item representations from the ratings dataset $\mathcal{R}$ using a CF approach. Then, the resulting *item embedding* maps each item $i$ to a vector representation $\mathbf{i}$ of its (latent) attributes. Note that these embeddings are typically not interpretable. Similarly, user preferences are captured by a *user embedding*, mapping users $u$ to a vector representation $\mathbf{u}$.

Methods including matrix factorization (Mnih & Salakhutdinov, 2007) or neural CF (Rendle et al., 2020; He et al., 2017; Beutel et al., 2018) are used to learn the user and item embeddings, which assumes a *two-tower model* (or *dual encoder*) in which users and items are passed through separate (but co-trained) deep neural nets (DNNs) to produce their respective vector embeddings $\mathbf{u}$ and $\mathbf{i}$. These are then combined via dot product to predict user-item affinity $\hat{r}_{i,u}$ (Yi et al., 2019; Yang et al., 2020). We view $\mathbf{i}$ as a (learned) latent feature vector characterizing item $i$ and $\mathbf{u}$ as parameterizing user $u$'s estimated *utility (or preference) function* over these features.

**Language Models (LMs).** In this work, we inject a user's behavioral information into a seq2seq LM (Vaswani et al., 2017) to generate personalized recommendation responses. We assume a dataset of the form $\mathcal{D} = \{(\mathbf{I}^{(k)}, \mathbf{i}^{(k)}, \mathbf{u}^{(k)}, Y^{(k)})\}_{k=1}^{|\mathcal{D}|}$, where $\mathbf{I}$ is a textual description of some item $i \in \mathcal{I}$ (e.g., descriptions, positive/negative reviews from different users); $\mathbf{i}$ is the CF embedding vector of $i$; $\mathbf{u}$ is the CF embedding vector of a user $u \in \mathcal{U}$; and finally, $Y$ is a textual response (e.g., compelling recommendation, endorsement or explanation) tailored to the user. We refer to Appendix C for details on the generation of $\mathcal{D}$.

Let $N_{\mathbf{I}}$ be an upper-bound on the length (number of tokens) of any item description $\mathbf{I}$.[1] The role of an LM is to predict the probability $\mathbb{P}(Y = \{y_n\}_{n=0}^{N-1} \mid y_0, \mathbf{I}, \mathbf{i}, \mathbf{u})$ of the personalized response $Y$ ($N$ tokens), conditioned on the item description $(\mathbf{I}, \mathbf{i})$ and user embedding $\mathbf{u}$.

In standard LMs, a Transformer (Wolf et al., 2019) architecture T encodes an item's textual context $\mathbf{I}$ as an $N_{\mathbf{I}}$-length sequence of embeddings $(z_0, \ldots, z_{N_{\mathbf{I}}-1})$ induced by the transformer's attention layers. For convenience, we concatenate these into a single embedding $z \in \mathcal{Z} \subseteq \mathbb{R}^d$, where $d$ is the dimension of the latent space. The text response $Y = \{y_n\}_{n=0}^{N-1}$ is sampled token-by-token in an auto-regressive manner using a decoder $\Psi$; i.e., $Y \sim \Psi(\cdot \mid z) := \prod_{n=0}^{N-1} \Psi(y_n \mid y_0, \ldots, y_{n-1}; z)$, where $y_0$ is a fixed start-of-sentence token (Chien & Kuo, 2019). To incorporate behavioral information into the LM, the standard LM is augmented with adapters (Pfeiffer et al., 2020) $W_I, W_U : \mathcal{V} \mapsto \mathcal{Z}$, to induce the language model: $\Psi \circ (\text{T} \times W_I \times W_U)$ (Jaech & Ostendorf, 2018). Here, T maps text-input tokens to $\mathcal{Z}$ whereas $W_I$ (resp., $W_U$) maps item (resp., user) CF-embedding vectors $\mathcal{V}$ to $\mathcal{Z}$. Importantly, T, $W_I$, and $W_U$ map tokens and CF vectors to a common space so that their relationship can be captured by the transformer's attention mechanism.

**Contextual Markov Decision Processes (CoMDPs).** CoMDPs have been used to model token-wise generative language problems (Li et al., 2016; Asadi & Williams, 2016; Jaques et al., 2019), and can also be used in conversational RSs. In this MDP, the LM acts as a policy which maps text inputs and user/item behavioral embedding vectors to generated responses.

Let $(\mathcal{C}, \mathcal{S}, \mathcal{A}, P, r, s_0, N)$ denote the CoMDP, where the observable context space $\mathcal{C}$ contains item/user information $\mathcal{I}, \mathbf{i}$ and $\mathbf{u}$. The horizon $N$ is the length of the generated text. The state space $\mathcal{S}$ at the $n$-th turn ($n < N$) is the sequence of tokens $\{y_0, \ldots, y_{n-1}\}$ generated thus far, with $s_0$ being the start-of-sentence token $y_0$. The action space $\mathcal{A}$ is the language token vocabulary, with action $a \in \mathcal{A}$ representing any possible next token. The transition kernel $P$ models the next token distribution given the current sequence and contexts, which coincides with the LM policy (and is thus known). Finally, the reward function $r$ measures the overall quality of the generated text. Our goal is to find a policy $\pi^*$ which achieves maximum expected cumulative return, i.e., $\pi^* \in \arg\max_\pi J_\pi := \mathbb{E}[\sum_{n=0}^{N-1} r_t \mid P, s_0, \mathcal{C}, \pi]$. Note that the size of the tokenized state and action spaces grow exponentially with the vocabulary size.

## 3 FACTUAL & PERSONALIZED RECOMMENDATIONS WITH LMS

A key question when using LMs for recommendation is how to effectively use the information captured by the RS embedding space to generate a factual, personalized, compelling, and relevant text response. Treating an LM as a factored distribution of item-user information over generated text tokens, one standard approach is to learn this model with *behavioral cloning (BC)* (Sasaki &

---

[1] If the actual description $\mathbf{I}$ has fewer tokens than $N_{\mathbf{I}}$, remaining spaces in the utterance will be padded by a specific token and masked.

Yamashina, 2020), by maximizing the conditional log-likelihood w.r.t. to the dataset $\mathcal{D}$:

$$\min_{\Psi} \ L_{\text{Cond}}(\Psi) := -\mathbb{E}_{(\mathbf{I},\mathbf{i},\mathbf{u},Y)\sim D}\big[\sum_{n=0}^{N-1} \log \Psi(y_n \mid y_0,\ldots,y_{n-1};\mathbf{I},\mathbf{i},\mathbf{u})\big].$$

While this model may learn to interpret the behavioral information captured in the RS embeddings, the LM might actually lean towards disregarding the embedding contexts due to the typically more predictable nature of token generation when given text inputs. Consequently, the model might concentrate solely on text information, effectively degenerating to a non-contextual LM. To prevent this from occurring, and more importantly to ensure the LM can offer a comprehensive RS experience, we incorporate four key metrics into our training procedure; namely, *personalization*, *precision*, *appeal*, and *preference relevance*. We detail these next.

**Precision.** LM-based personalized recommendation can be viewed as a special form of abstractive summarization (Zhang et al., 2020a; Liu et al., 2022): the generated text should capture item characteristics that explain why a user would benefit from the recommendation. To preserve the RS's integrity, of course, one must emphasize *truthfulness* in its recommendation. That is, the RS's generated recommendation should describes genuine merits (and drawbacks) of the item, rather than persuasive distortions.

While recent summarization techniques produce highly coherent texts, they often suffer from *hallucinations* (Ji et al., 2023) – the tendency to generate information unsupported by the input text. Such factual inconsistencies may therefore limit their real-world applicability. Inspired by Roit et al. (2023) and Honovich et al. (2022), we evaluate factuality in our LM-based RS using an *entailment reward* (Bowman et al., 2015). Unlike widely-used metrics, such as ROUGE (Lin, 2004), that are ineffective at hallucination detection, we adopt a *textual entailment* (or natural language inference (NLI)) metric to measure truthfulness of our generated text, viewing it as a partial summary of an items's description. Particularly, given a description $\mathbf{I}$, we define the NLI score $\text{NLI}(Y;\mathbf{I})$ of text-token sequence $Y$ as the probability of entailment under a classifier trained on several textual entailment datasets (see e.g., MacCartney & Manning (2007)). While this metric is not specifically tailored to summarization tasks, Honovich et al. (2021) show that it effectively detects factual inconsistencies in generated text. Since faithful summaries should be textually entailed by the input documents, such a metric provides informative feedback about the precision of generated item texts.

Of course, factual entailment is clearly insufficient in and of itself. In fact, it is rather easy to optimize a degenerate response which maximizes factual entailment (e.g., producing summaries that are highly extractive (Ladhak et al., 2021) or uninformative (Skalse et al., 2022)). In what follows we describe three other metrics we require for a comprehensive recommendation experience.

**Appeal.** Recent work has paid increasing attention to enriching recommendations to appeal to users (Felfernig et al., 2007; Zhang et al., 2020b). To the extent that we do not sacrifice user welfare, personalization, or factuality, such recommendations have value as they encourage users to accept recommendations of high personal utility. With recent LM technologies (Google et al., 2023; OpenAI, 2023), a plausible approach is to simply prompt an LM to generate an *endorsement* to complement its item recommendation. Such an endorsement, apart from being factual, should be compelling for the user. However, without systematic evaluation of such methods (e.g., do users find them appealing or compelling), it remains unclear whether they can improve the user experience. Quantifying appeal is challenging, as it may depend on subjective factors such as *style* (concise phrases over detailed explanations) and *language* (compelling, eloquent pitches over dry factual summaries).

To assess appeal, we use a dataset of pairwise human/machine demonstrations (see Appendix C for details on its construction). We develop an *appeal model* which scores the generated text $Y$ and assess how compelling it are, using learning from human/AI feedback (LAIF) (Christiano et al., 2017). Specifically, let $\mathcal{D}_{\text{app}} = \{(Y_w^{(k)}, Y_l^{(k)};\mathbf{I})\}_{k=1}^{|\mathcal{D}_{\text{app}}|}$ be a labeled dataset reflecting the relative appeal of two recommendation texts $Y_w, Y_l$ given textual item description $\mathbf{I}$. Here, $Y_w \succ Y_l|\mathbf{I}$ indicates that $Y_w$ is more compelling given $\mathbf{I}$. Assuming these relationships are governed by a latent model $\text{App}(Y;\mathbf{I})$, we parameterize it via Bradley-Terry (Huang et al., 2006), where the appeal distribution is defined by

$$p_{\text{app}}(Y_w \succ Y_l;\mathbf{I}) = \frac{\exp(\text{App}(Y_l;\mathbf{I}))}{\exp(\text{App}(Y_w;\mathbf{I})) + \exp(\text{App}(Y_l;\mathbf{I}))}.$$

We estimate the parameters of the reward model via maximum likelihood by formulating the problem as a binary classification with a negative log-likelihood loss: $L_{\text{MLE}}(\text{App}, \mathcal{D}_{\text{app}}) =$

$-\mathbb{E}_{(Y_w,Y_l;\mathbf{I})\sim\mathcal{D}_{\text{app}}}\log\sigma(\text{App}(Y_w;\mathbf{I})-\text{App}(Y_l;\mathbf{I}))$. To reduce variance, we normalize this by subtracting the population mean so that $\mathbb{E}_{(Y,\mathbf{I})\sim\mathcal{D}_{\text{app}}}[\text{App}(Y;\mathbf{I})]=0$ for all contexts $\mathbf{I}$.

**Personalization.** A conversational RS is only effective to the extent that it recommends, and ultimately, the user accepts, items of significant value to the user. Thus, *personalization* is perhaps the foremost criterion with which to evaluate an LM-based RS. Particularly, we wish to evaluate the extent to which the LM's generated response $Y$ corresponds to an item with high utility for a user $u$. To this end, we develop a scoring model $\text{Per}(Y;\mathbf{i},\mathbf{u})$ which interprets the semantics of text $Y$ to quantify its value as a personalized recommendation.

To achieve this, recall the dataset $\mathcal{D}=\{(\mathbf{I}^{(k)},\mathbf{i}^{(k)},\mathbf{u}^{(k)},Y^{(k)})\}_{k=1}^{|\mathcal{D}|}$ of item description, item CF embedding vector, user CF embedding vector, and textual response tailored to the user, and the estimated utility that is the dot product $\hat{r}=\mathbf{i}\cdot\mathbf{u}$ of their CF embedding vectors. To measure personalization one could learn a reward model $\text{Per}(Y;\mathbf{i},\mathbf{u})$ that predicts the utility $\hat{r}$ based on textual response $Y$. However, this approach relies on a strong assumption that such text alone is predictive of user-item utility. Alternatively, we can also employ the LAIF approach (Christiano et al., 2017) that leverages preference feedback to learn a personalization reward model. Using the same dataset $\mathcal{D}$, and assuming the recommendation text is more personalized than item description, i.e., $Y\succ\mathbf{I}|\mathbf{i},\mathbf{u}$,[2] a Bradley-Terry based personalization reward model $\text{Per}(Y;\mathbf{i},\mathbf{u})$ can be learned by minimizing the negative log-likelihood loss: $L_{\text{MLE}}(\text{Per},\mathcal{D}_{\text{per}})=-\mathbb{E}_{(Y,\mathbf{I};\mathbf{i},\mathbf{u})\sim\mathcal{D}_{\text{per}}}\log\sigma(\text{Per}(Y;\mathbf{i},\mathbf{u})-\text{Per}(\mathbf{I};\mathbf{i},\mathbf{u}))$.

**Preference Relevance.** While appeal and personalization distinguish compelling recommendations for a user from simple factual item summaries, they do not capture the full *relevance* of the LM's response w.r.t. a user's preferences. For example, the LM might still describe item attributes that the user has no interest in (positively or negatively). To address this, we assume access to a *textual description* of a user's preferences (we later describe how we create these from user CF embeddings). We train an additional reward model, $\text{Prel}(Y;\mathbf{I},\mathbf{u})$, which explicitly measures the semantic similarity between a user's description of preferences and the generated text, constrained to attributes of the recommended item. More specifically, we assume availability of a mapping from a user's CF embedding vector $\mathbf{u}$ to a textual description of their preferences. We train this mapping using a dataset of user embeddings and textual descriptions $\{\text{U}_j(\mathbf{u})\}_{j=1}^J$ (see Appendix C for details on the generation of this dataset).

Next, for each $(\mathbf{I},\mathbf{u},Y)$, we encode the user's textual preferences $\{\text{U}_j(\mathbf{u})\}_{j=1}^J$ and the item description $\mathbf{I}$ using an *LM semantic encoder*.[3] Then, we rank each textual preference using cosine similarity of its encoded counterpart and encoded item. This, in turn, determines which of the $J$ preference texts are most relevant to the item of interest. Finally, we use the same model to encode the recommendation response $Y$ and compute its cosine similarity with the user preference texts.

We define the *preference relevance* score $s$ of $Y$ w.r.t. user-item pair $(\mathbf{u},\mathbf{i})$ to be the average of the above cosine similarity scores. To this end, we train the reward model $\text{Prel}(Y;\mathbf{I},\mathbf{u})$ by minimizing an $\ell_2$ regression loss $L_{\text{REG}}(\text{Prel},\mathcal{D}_{\text{Prel}})=\mathbb{E}_{(\mathbf{I},\mathbf{u},Y,s)\sim\mathcal{D}_{\text{Prel}}}(s-\text{Prel}(Y;\mathbf{I},\mathbf{u}))^2$.

## 4 Reinforcement Learning based Fine-tuning

RL from AI feedback (RLAIF) can effectively align LMs to metrics that are labeled by off-the-shelf LMs in lieu of humans. Recent work (Lee et al., 2023; Bai et al., 2022; Zhu et al., 2023) has shown that hybrid human-AI preference models, together with *self-improving* fine-tuning, outperforms traditional supervised fine-tuned baselines and offers additional benefits relative to standalone RL fine-tuning with human feedback (RLHF). Using the four principles for LM-based recommendation outlined in Section 3, we develop four reward models to help train and evaluate LM w.r.t. personalization, precision, appeal and preference relevance. We then devise an RLAIF technique to fine-tune an LM with a joint reward model defined by these four components.

In multi-objective RL, it is common to aggregate reward models via *linear scalarization* (Peschl et al., 2021) (which corresponds to solving for an optimum on the convex Pareto frontier). Given a text

---

[2]Instead of comparing the recommendation text with item description, one could instead construct a dataset with two texts and a labeled rating order (see Appendix C for details).

[3]Much like *Sentence-T5* (Ni et al., 2022a) and *T5-based Retrievers* (Ni et al., 2022b), the semantic encoder $E$ maps textual inputs (e.g., item description $\mathbf{I}$ or user preference texts $\{\text{U}_j(\mathbf{u})\}_{j=1}^J$) to a latent space in $\mathbb{R}^{d_{\text{enc}}}$.

response $Y = \{y_n\}_{n=0}^{N-1}$, item description $\mathbf{I}$, and user-item CF embedding vectors $(\mathbf{u}, \mathbf{i})$, we define the LM-based RS reward recommender reward by:

$$r(y_n; y_{0:n-1}; \mathbf{I}, \mathbf{i}, \mathbf{u}) = \begin{cases} \eta_1 \mathrm{NLI}(Y; \mathbf{I}) + \eta_2 \mathrm{App}(Y; \mathbf{I}) + \eta_3 \mathrm{Per}(Y; \mathbf{i}, \mathbf{u}) + \eta_4 \mathrm{Prel}(Y; \mathbf{I}, \mathbf{u}) & \text{if } y_n = [\mathrm{EOS}]; \\ 0 & \text{otherwise,} \end{cases}$$

where $\eta_1, \eta_2, \eta_3, \eta_4 \geq 0$ are importance weights for the component rewards, and are treated as hyper-parameters (optimized using e.g., grid search).

Recall the LM $\mathbb{P}_\theta(Y \mid y_0; \mathbf{I}, \mathbf{i}, \mathbf{u})$ with item text $\mathbf{I}$, item-user CF embedding vectors $(\mathbf{i}, \mathbf{u})$ and the reward model $r(Y, \mathbf{I}, \mathbf{i}, \mathbf{u})$, which jointly measures appeal, factuality, preference-relevance, and personalization of a recommendation response. The goal in LM fine-tuning is to maximize the average overall quality of the generated text, i.e., $\max_\theta \ \mathbb{E}_{(\mathbf{I}, \mathbf{i}, \mathbf{u})} \mathbb{E}_{\mathbb{P}_\theta(Y|\mathbf{I}, \mathbf{i}, \mathbf{u})}[r(Y; \mathbf{I}, \mathbf{i}, \mathbf{u})]$. Using the CoMDP framework, it is easily shown that this learning problem can be solved with on-policy REINFORCE (Williams, 1992), in which the policy gradient is estimated using trajectories generated by the current LM policy.

A risk of RL fine-tuning based on an AI-feedback is that it might overfit to the model, thereby degrading the "skill" of the original LM. To alleviate this, we add a KL regularization term (Ouyang et al., 2022; Stiennon et al., 2020) between the LM $\mathbb{P}_\theta(Y|\mathbf{I}, \mathbf{i}, \mathbf{u})$ and the pre-trained model $\mathbb{P}_{\mathrm{pre}}(Y|\mathbf{I}, \mathbf{i}, \mathbf{u})$ to the CoMDP objective function. Leveraging the auto-regressive nature of LMs, KL regularization is applied over the entire MDP trajectory, reducing the objective function to

$$\max_\theta \ J(\theta) := \mathbb{E}_{(\mathbf{I}, \mathbf{i}, \mathbf{u})} \mathbb{E}_{\mathbb{P}_\theta(Y|\mathbf{I}, \mathbf{i}, \mathbf{u})} \left[ r(Y; \mathbf{I}, \mathbf{i}, \mathbf{u}) - \beta \log \frac{\mathbb{P}_\theta(Y|\mathbf{I}, \mathbf{i}, \mathbf{u})}{\mathbb{P}_{\mathrm{pre}}(Y|\mathbf{I}, \mathbf{i}, \mathbf{u})} \right]. \tag{1}$$

This is equivalent to a KL-regularized CoMDP. The LM policy $\pi_\theta$, where $\mathbb{P}_\theta(Y|\mathbf{I}, \mathbf{i}, \mathbf{u}) = \prod_{n=0}^{N-1} \pi_\theta(s_n|a_n; c)$, can be learned by computing the policy gradient of the KL-regularized objective online, or by employing an off-policy RL algorithm, e.g., SAC (Haarnoja et al., 2018), in-sample softmax (Xiao et al., 2023), CQL (Kumar et al., 2020), that leverages offline data $\mathcal{D}$ for more efficient training. (See Appendix D for full exposition of these algorithms.) KL regularization, intended to avoid over-fitting to the reward model, can also alleviate out-of-distribution generalization issues common in offline RL (Kumar et al., 2019).

## 5 EXPERIMENTS

We conduct empirical validations of P⁴LM, focusing on assessing its capability to generate factual, personalized, and compelling recommendation endorsements. We examine the hypothesis that the reward models (RM) detailed in Section 3 significantly increase the personalization, precision, appeal and preference relevance of movie recommendations. We use the MovieLens 25M recommendation dataset (Harper & Konstan, 2015), which contains ratings of $62,423$ movies by $162,541$ users. We use these movie-user interactions to generate movie descriptions, user-preference texts, and sample recommendation responses by prompting a PaLM2-L LM (Google et al., 2023); our data generation procedures are detailed in Appendix C. The resulting datasets have four components: (1) movie descriptions $\mathbf{I}$, (2) item-user behavioral embeddings $(\mathbf{i}, \mathbf{u})$, (3) user preference texts $\mathrm{U}(\mathbf{u})$, and (4) sample responses $Y$. We experiment with a set of LMs in the PaLM2 family (Google et al., 2023). To incorporate user and movie embedding vectors into the LM (Section 3) we construct LMs by augmenting these LMs with adapter layers. Specifically, we train two models, P⁴LM and P⁴LM-S, derived from PaLM2-XS and PaLM2-XXS, respectively. Our reward mixing weights, optimized using grid search, are $(\eta_1, \eta_2, \eta_3, \eta_4) = (2.0, 0.1, 1.0, 1.0)$.

To demonstrate the efficacy of our models P⁴LM and P⁴LM-S, we compare them with the following SOTA baselines on our conversational movie recommendation task: (i) **PaLM2-L**, a pre-trained model prompted using movie descriptions, user preference texts and instructions to generate a response that respects our four recommender principles; (ii) **Supervised Fine-Tuned with Text (SFT-Text)**, a PaLM2-XS model fine-tuned with the dataset above, with explicit user-item texts as input; (iii) **Supervised Fine-Tuned (SFT)**, a PaLM2-XS model fine-tuned to use user-item embedding vectors.

In Section 5.1, we first validate the efficacy of the RMs using AI-generated examples with known labels. In the following sections, we measure the performance of our approach via *model-based* and *human* evaluation.

Table 2: Model-based Evaluation Based on the Principles of Recommendation LM.

| Method | Precision | Personalization | Appeal | Pref. Relevance |
|--------|-----------|-----------------|--------|-----------------|
| PaLM2-L | $0.52 \pm 0.03$ | $-0.04 \pm 0.04$ | $0.36 \pm 0.04$ | - |
| SFT-Text | $0.58 \pm 0.03$ | $0.10 \pm 0.04$ | $0.46 \pm 0.03$ | $\mathbf{-1.01 \pm 0.06}$ |
| SFT | $0.58 \pm 0.03$ | $-0.15 \pm 0.04$ | $0.33 \pm 0.04$ | $-1.08 \pm 0.07$ |
| P$^4$LM | $\mathbf{0.72 \pm 0.03}$ | $\mathbf{0.23 \pm 0.04}$ | $0.63 \pm 0.04$ | $-1.18 \pm 0.07$ |
| P$^4$LM-S | $0.65 \pm 0.03$ | $0.18 \pm 0.04$ | $\mathbf{0.72 \pm 0.04}$ | $-1.10 \pm 0.08$ |

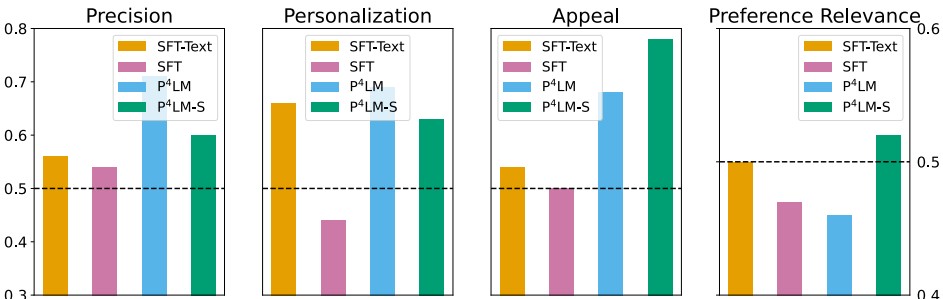

Figure 2: Win Rates of Different Model-based Scores against PaLM2-L

## 5.1 VALIDATION OF REWARD MODELS

It is crucial to assess if the scores of the RMs reflect human values. To test this, we prompt an LM with $(\mathbf{I}, U(\mathbf{u}))$, generating two distinct-quality endorsements. For instance for NLI, the LM produces one response adhering to the item description and another with added hallucinations. The first response is more factually accurate. With $n$ examples, we evaluate the NLI RM's accuracy in distinguishing better from worse endorsements in precision. This process is repeated for App and Per RMs, with accuracies presented in Table 1. For robustness in evaluation, we use GPT-4 (OpenAI, 2023) for generating the test data. See Appendix A.1 for additional details.

Table 1: The accuracies of the RMs on GPT-4 generated examples.

| RM | NLI | App | Per |
|----|-----|-----|-----|
| Accuracy | 1.0 | 1.0 | 0.9375 |

## 5.2 MODEL-BASED AND HUMAN EVALUATION

**Model-Based Evaluation**  We conduct *model-based* evaluations using the four criteria and their corresponding RMs from Section 3, namely, NLI, App, Per, and Prel. Specifically, we report the scores of responses $Y$ from different models on a held-out, unlabeled dataset $\mathcal{D}_{\text{test}}$ consisting of 96 non-overlapping user-movie pairs. We also assess the relative improvement of each LM over the PaLM2-L common baseline that we use to generate the supervised learning dataset. We do this by computing the (a) *win rate* (number of occurrences on which a candidate LM outperforms PaLM2-L), (b) *absolute increase* (the magnitude of the score improvement), and (c) *percentage increase* (the relative score improvement).[4]

Our results in Table 2 highlight the robust performance of P$^4$LM in three pivotal dimensions: Precision, Personalization, and Appeal. P$^4$LM attains the highest precision score by a wide margin, underscoring its ability to mitigating the risks of misleading users with hallucinated information about recommended items. It also outperforms on personalization and appeal. It appears that P$^4$LM compromises on preference relevance to achieve these gains, with qualitative comparisons (see Appendix A.2 for details) on the texts generated by P$^4$LM and SFT verifying these phenomenons. We believe that personalization is by far the most important aspect of recommendation quality, while precision/factuality is the most critical property of any endorsement text.

Figure 2 displays the win rates of various LMs compared to PaLM2-L.[5] Notably, SFT and SFT-Text show low precision scores, indicating a tendency to overfit and generate inaccurate movie details.

---

[4]Precise definitions of these relative metrics are provided in Appendix B.

[5]See Appendix A for detailed absolute and percentage increases in Figures 4 and 3.

Table 3: Human Evaluation Results: Ratio of Preferred Outputs over PaLM2-L Outputs

| Method | Precision | Personalization | Appeal | Pref. Relevance |
|--------|-----------|-----------------|--------|-----------------|
| SFT-Text | 0.29 | 0.458 | 0.375 | 0.458 |
| SFT | 0.458 | **0.583** | 0.458 | 0.583 |
| P$^4$LM | **0.5** | **0.583** | **0.667** | **0.708** |
| P$^4$LM-S | **0.5** | 0.5 | 0.5 | 0.333 |

Table 4: Model-based Evaluation Scores using a Single Reward Model (Ablation Studies)

| Method | Precision | Personalization | Appeal | Pref. Relevance |
|--------|-----------|-----------------|--------|-----------------|
| NLI | **0.77 ± 0.02** | −0.14 ± 0.04 | 0.43 ± 0.04 | −1.08 ± 0.07 |
| Personalization | 0.55 ± 0.03 | **0.39 ± 0.04** | **0.94 ± 0.04** | −1.11 ± 0.07 |
| Appeal | 0.56 ± 0.03 | 0.14 ± 0.04 | 0.76 ± 0.04 | −1.06 ± 0.06 |
| Pref. Relevance | 0.51 ± 0.03 | −0.07 ± 0.04 | 0.51 ± 0.04 | **−1.02 ± 0.07** |

P$^4$LM's higher personalization score, even surpassing SFT-Text that directly uses user profile texts, highlights its superior utilization of user behavioral information in the user embedding vector. Unlike SFT, P$^4$LM effectively leverages this data, emphasizing the advantage of RL-training.

**Human Evaluation**   Table 3 shows our human evaluation results. We presented *raters* with two endorsement texts: one from PaLM2-L and another by a different model, to assess their relative performance. Raters evaluated these texts based on four criteria aligned with our model-based evaluation. For more information, refer to Appendix E.

In precision, P$^4$LM and P$^4$LM-S equaled PaLM2-L in raters' preferences, despite their smaller sizes (XS and XXS). P$^4$LM also outperformed baseline models in personalization, appeal, and preference relevance. Interestingly, human evaluation favored SFT over SFT-Text, contrary to the model-based results. This may be because SFT-Text, sharing the same inputs as PaLM2-L and trained on its generated dataset, is limited by PaLM2-L's capabilities. Conversely, SFT's use of user embeddings enhances personalization, giving it an edge over SFT-Text. Overall, the human evaluation further corroborates the results of the model-based assessment, highlighting P$^4$LM's superior performance.

**Ablation Studies**   Our ablation studies, outlined in Table 4, show that training with a single RM predictably results in policies scoring the highest in the model-based evaluation for the specific RM targeted (see Precision, Personalization, Preference Relevance scores). Intriguingly, a model trained solely on Personalization not only excels on that metric, but also attained the highest score in Appeal, suggesting a possible correlation where recommendation text that is well-tailored to a user's preferences may be inherently appealing. We also explore the impact of varying the mixing-weight combination, which is discussed in Appendix A.

We have conducted human evaluations on models trained with individual RMs, with results being detailed in Table 5 in Appendix A. Here, we briefly highlight key observations, directing readers to the appendix for in-depth discussion. Notably, the performance ranking of these models in human evaluations diverges from those in Table 4. The model trained solely with the NLI reward, expected to excel in Precision, surprisingly scored the lowest in human evaluations. This suggests potential *reward hacking*, where the policy learner exploits the single RM to inflate scores. This underscores the necessity of using multiple RMs, as each serves as a regularizer thwarting over-optimization of a single RM, ensuring balanced performance. Our ablation studies reveal that focusing on a particular RM boosts its model-based score, but this trend is not always reflected in human evaluations, further indicating the possibility of reward hacking. This stresses the importance of adopting a diverse set of RMs in RLAIF to counteract such issues.

# 6   RELATED WORK

Our work intersects multiple areas of research, notably personalized recommendation systems, leveraging of language models (LMs) and reinforcement learning, recommendation integrity.

**Personalized Recommender Systems**   Recommender systems have ubiquitous applications permeating e-commerce, content providers, social media, etc., with collaborative filtering (CF) (Schafer et al., 2007) as the prominent modeling technique. Early works include matrix factorization approaches (Mnih & Salakhutdinov, 2007), which became a foundation for subsequent deep learning methods like neural CF (He et al., 2017). Notably, dual encoder architectures emerged, where user and item embeddings are co-trained (Yi et al., 2019; Yang et al., 2020). While traditional CF approaches worked well in many applications, advances in deep personalization allow user and item embeddings to capture more nuanced preferences (Rendle et al., 2020; Beutel et al., 2018).

**Conversational Recommender Systems & Language Models**   Conversational recommender systems (RSs) add an interactive layer over traditional RSs with an conversational agent interacting with users, understanding their preferences and refining recommendations through dialogue (Chen et al., 2019; Zhou et al., 2020; Lei et al., 2020; Li et al., 2018; Sun & Zhang, 2018; Christakopoulou et al., 2016). This paradigm integrates aspects of natural language understanding, making it ripe for integrating LMs. Leveraging language models in RSs is a relatively recent development. With the advance of transformer architectures (Vaswani et al., 2017; Wolf et al., 2019), LMs have found use-cases beyond typical NLP tasks. Researchers began exploring the synthesis of textual data with user preferences to enhance the personalization and expressiveness of RSs (Jaech & Ostendorf, 2018; Xia et al., 2023). Our work situates itself in this space, but with an added twist: we aim to generate compelling narratives that genuinely communicate the relevance of a recommendation.

**Transparency and Truthfulness in Recommendation Systems**   Maintaining integrity in RSs is technically challenging yet critically important. The potential that RS algorithms inadvertently mislead users or reinforce biases has been highlighted (Abdollahpouri et al., 2019; Shen et al., 2023; Cabello et al., 2023). Therefore, increasingly researchers are not only prioritizing the recommendation efficacy but also the fairness, transparency, and interpretability of RS algorithms (Beutel et al., 2019; Ghazimatin et al., 2020; Chen et al., 2023). Our work takes cues from this domain, emphasizing truthful and precise recommendations that articulate genuine merits rather than compelling distortions.

**Reinforcement Learning with Human/AI Feedback**   The integration of reinforcement learning (RL) with language models has emerged as a compelling strategy for refining model behavior beyond supervised fine-tuning (Williams, 1992; Ranzato et al., 2016). The RL with Human Feedback (RLHF) methodology (Christiano et al., 2017; Bai et al., 2022), in particular, has gained traction, where model responses are ranked by human evaluators and subsequently used to fine-tune models through techniques like Proximal Policy Optimization (Schulman et al., 2017). In a different vein, Inverse Reinforcement Learning (Abbeel & Ng, 2004) has been employed to extract objectives from expert demonstrations in textual settings (Daniels-Koch & Freedman, 2022; Sun, 2023). Additionally, there's a growing interest in AI-driven feedback mechanisms, where preferences are labeled by off-the-shelf LMs in lieu of humans (Lee et al., 2023; Bai et al., 2022). These endeavors underline the potential of using RL to steer LMs towards better alignment with human preferences and nuanced task objectives.

## 7   CONCLUSION

We studied language modeling for personalized recommendation. By developing novel reward models which quantify prominent attributes of personalized recommendations, one may develop self-improving LM methodologies via reinforcement learning with AI feedback. As a result, our developed LM; namely P[4]LM, not only parses language semantics, but also understands latent user preferences (encoded in the CF embedding space). P[4]LM provides factual, compelling, personalized endorsement of relevant items, connecting the items with users' preferences, thereby increasing the likelihood of users accepting high-value recommendations.

We demonstrated the efficacy of P[4]LM on the MovieLens 25M dataset. Particularly, our agent better understands user behaviors encoded in the CF embedding space and delivers precise, compelling, personalized movie recommendation narratives. Our work is a step toward creating intelligent conversational recommenders which can compellingly explain the intricacies between item features and user preferences. Future work includes (i) improving P[4]LM's capabilities to generate longer responses beyond standard single-shot autoregressive decoding; (ii) extending our RL fine-tuning approach to handle multi-turn conversational recommendations; (iii) developing better reasoning capabilities to trade off between user-item preferences and constraints; (iv) and expanding the LM's functionality beyond recommendation, to also include technical support, negotiations, etc.

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
