---

**Prompt to GPT-4 for Evaluating the NLI RM**

Craft two endorsements of the movie {title} whose plot is provided below. One of the two endorsements should be more factually consistent, relying solely on the information from the plot. The other one should contain extra information not provided in the plot. That is, these two endorsements should have different qualities in terms of their precision. Avoid copying the plot verbatim. It should be easy to tell the difference. The lengths of the two endorsements should be similar.
Here is the plot: {plot}
Generate the two endorsement texts. Firstly generate a better one and then generate a worse one.

---

**Prompt to GPT-4 for Evaluating the Appeal RM**

Craft two endorsements of the movie titled '{title}' whose plot is provided below. The endorsements should be factual, relying solely on the information from the plot. Avoid copying the plot verbatim. Ensure each endorsement is directly tied to the plot elements. Do not ever start an endorsement with lead-in sentences that are not relevant to the movie plot. These two endorsements should have different qualities in terms of their compellingness. That is, one endorsement should be more compelling than the other one, and it should be easy to tell the difference. The lengths of the two endorsements should be similar.
Here is the plot: {plot}
Generate the two endorsement texts. Firstly generate a worse one and then generate a better one.

---

**Prompt to GPT-4 for Evaluating the Personalization RM**

Craft two personalized endorsements of the movie titled '{title}' whose plot is provided below. The user's preferences are also summarized below. Your endorsements should be factual, relying solely on the information from the plot. Avoid copying the plot verbatim. Tailor an endorsement to align with the user's preference profile, highlighting aspects of the movie that would particularly appeal to them. Ensure each appeal is directly tied to the plot elements. Do not ever start an endorsement with lead-in sentences that are not relevant to the movie plot or user preferences. These two endorsements should have different qualities in terms of how well they are personalized to the user preference profile. That is, one endorsement should be better personalized than the other one, and it should be easy to tell the difference. The lengths of the two endorsements should be similar.
Here is the plot: {plot}
Here is the user preference profile: {user_profile}
Generate the two endorsement texts. Firstly generate a worse one and then generate a better one.

## A  ADDITIONAL RESULTS

### A.1  VALIDATION OF REWARD MODELS

In the development of our factual and personalized recommender Language Model (LM), we employed four distinct Reward Models (RMs). The efficacy of these RMs is critical, as they need to accurately reflect human values and preferences in the context of endorsement text generation.

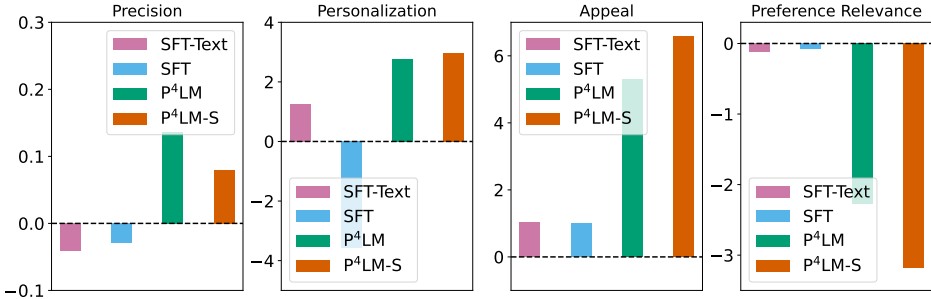

Figure 3: The absolute model-based score increases compared against PaLM2-L.

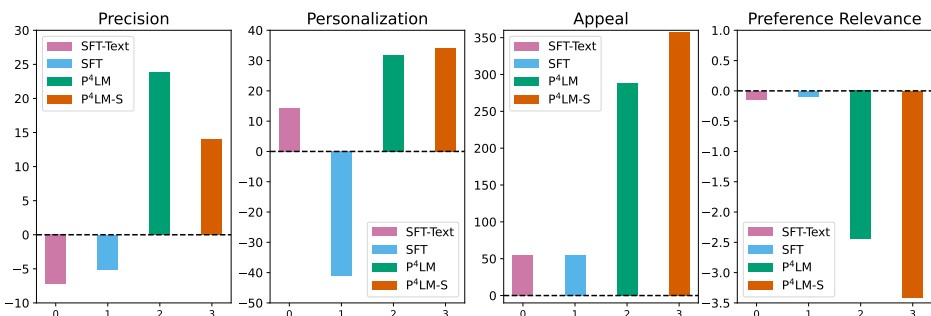

Figure 4: The percentage increases in model-based score compared against PaLM2-L.

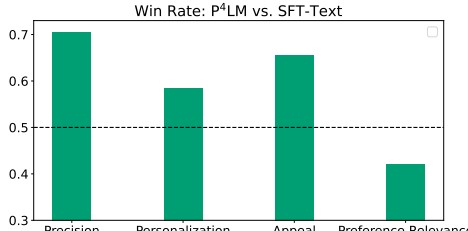

Figure 5: Win rate of $P^4$LM over SFT-Text.

The core objective of this evaluation is to determine whether the RMs can reliably distinguish between varying qualities of endorsement texts in terms of factual consistency, personalization, and appeal, respectively. This involves assessing if a specific RM, such as the NLI RM for factual consistency or the Appeal RM for textual appeal, can correctly score texts in alignment with their actual quality.

To validate the effectiveness of these RMs, we generated synthetic examples using GPT-4 OpenAI (2023). Our approach involved arbitrarily selecting 32 pairs of movies and users. For each pair, GPT-4 was prompted to generate two endorsement texts. These texts were designed to have distinctly different qualities in terms of a specific measure (e.g., one text being more compelling or personalized than the other, or one being more factually consistent). See the previous page for the prompts we used for the generation of these texts.

The evaluation criteria were straightforward: for each pair of texts, we already knew which one was superior based on the targeted measure. The task for the RMs was to correctly classify these texts — identifying the one with higher quality in the context of the specific measure being evaluated. The accuracy of the RMs in correctly classifying the superior text was calculated, providing a clear metric of their effectiveness. The outcomes of this evaluation are presented in Table 5.

**Model-based Evaluation**    Figure 3 and Figure 4 elucidate the absolute and percentage increases of each method compared to our common baseline, PaLM2-L, respectively. In correlation with the

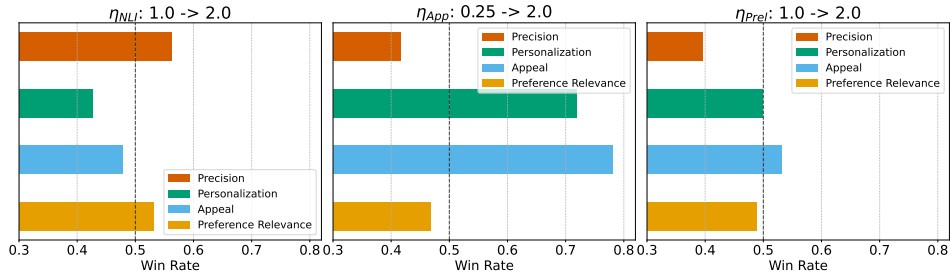

Figure 6: Win Rates While Changing the Mixing Weights of Reward Models.

Table 6: Model-based Evaluation Scores using a Single Reward Model (Same as Table 4)

| Method | Precision | Personalization | Appeal | Pref. Relevance |
|---|---|---|---|---|
| NLI | $\mathbf{0.77 \pm 0.02}$ | $-0.14 \pm 0.04$ | $0.43 \pm 0.04$ | $-1.08 \pm 0.07$ |
| Personalization | $0.55 \pm 0.03$ | $\mathbf{0.39 \pm 0.04}$ | $\mathbf{0.94 \pm 0.04}$ | $-1.11 \pm 0.07$ |
| Appeal | $0.56 \pm 0.03$ | $0.14 \pm 0.04$ | $0.76 \pm 0.04$ | $-1.06 \pm 0.06$ |
| Pref. Relevance | $0.51 \pm 0.03$ | $-0.07 \pm 0.04$ | $0.51 \pm 0.04$ | $\mathbf{-1.02 \pm 0.07}$ |

Table 7: Human Evaluation Scores using a Single Reward Model (Ablation Studies)

| RM | Precision | Personalization | Appeal |
|---|---|---|---|
| NLI | $4.18 \pm 0.05$ | $3.92 \pm 0.05$ | $4.36 \pm 0.06$ |
| Personalization | $4.33 \pm 0.05$ | $3.96 \pm 0.06$ | $4.05 \pm 0.06$ |
| Appeal | $4.53 \pm 0.05$ | $3.94 \pm 0.06$ | $4.43 \pm 0.05$ |
| Pref. Relevance | $4.44 \pm 0.06$ | $3.93 \pm 0.07$ | $4.39 \pm 0.06$ |

observations highlighted in Table 2, P[4]LM exhibits superior performance, reflected in elevated scores across Factuality, User Preference, and Appeal metrics. The observed percentage surges in these metrics surpass 10%, with the Appeal score witnessing approximately a 300% boost. However, P[4]LM does experience marginal diminutions in scores and win rates for Preference Relevance, as also shown in Section 5, marking a reduction by approximately -2 ∼ -3%.

Regarding the comparison between SFT and SFT-Text, both methods exhibit comparable performance across various metrics, barring the Personalization score. This divergence suggests that the nature of our task is inherently more intricate, due to the implicit reliance on user behavioral embedding vectors, rather than the straightforward utilization of text inputs. Specifically, SFT requires a meticulous extraction and interpretation of user preference information from the behavioral embedding vector to generate personalized recommendation endorsements effectively. In contrast, SFT-Text, utilizing user profile text as direct input, can generate user-aligned outputs more intuitively, as it has the flexibility to allocate its attention to specific user profiles selectively.

However, the dependency on text inputs introduces its own limitations, possibly omitting subtle user preference behaviors that can be captured more effectively through user embeddings, derived from the historical interactions of the users. This difference in capturing user preferences signifies that a singular approach focusing on supervised learning is suboptimal for deciphering and utilizing the intricate user preference information ingrained in embeddings. The notable improvement in user preference scores by P[4]LM, compared to baselines, highlights the effectiveness of our approach in addressing this limitation, emphasizing the substantial benefits of integrating LMs and RL for generating nuanced, factual, and personalized recommendations.

Figure 5 shows the comparison between P[4]LM and SFT-Text and illustrates the advantages of our approach. P[4]LM's high win rates in the first three metrics highlight its proficiency in effectively utilizing RS embeddings, a contrast to text-based methods, allowing for a more nuanced capture and representation of user preferences. This distinction serves to spotlight the potential of embedding-centric approaches in advancing personalized recommendation generation.

**Ablations** Our ablation studies, as outlined in Table 6, demonstrate a predictable yet informative trend: when a single RM is optimized during training, the resulting policy predominantly excels in that specific metric (notably in Precision, Personalization, and Preference Relevance scores). An intriguing observation is the performance of the model trained solely on Personalization. This model not only excels in its targeted metric but also achieves the highest score in Appeal. This suggests a potential correlation: recommendation texts that are finely tailored to a user's preferences might inherently be more appealing.

Figure 6 shows the win rate of policies trained with specific reward mixing weights, compared to a base policy trained with another reward mixing weights. The left figure illustrates the scenario when the NLI reward weight is increased from 1.0 to 2.0. Here, we computed the ratio of instances where the policy with an NLI weight of 2.0 achieved higher RM scores than the one with 1.0 by analyzing 96 pairs of (movie, user) and their corresponding endorsement texts. This win rate was calculated for all four RMs: Precision, Personalization, Appeal, and Preference Relevance, offering a nuanced view of how altering a single RM weight influences the overall policy performance across different metrics.

Further exploration into the impact of varying the mixing-weight combinations, presented in Figure 6, reveals several noteworthy trends:

1. Increasing the focus on Appeal positively influences Personalization, suggesting a symbiotic relationship between these two metrics.

2. Enhancing certain metrics often results in a decrease in others, highlighting the inherent trade-offs in our multi-objective optimization problem.

3. Emphasizing NLI slightly increases the Preference Relevance score, though not significantly.

4. This correlation appears to be asymmetric. That is, amplifying Preference Relevance degrades Precision.

Table 7 presents the results from human rater evaluations, which unveil a significant discrepancy between model-based and human assessments. Specifically, the model trained exclusively with NLI reward, while logically anticipated to excel in Precision, recorded the lowest score in human evaluations. This points to potential 'reward hacking,' where the policy learner exploits the single RM to achieve artificially elevated scores. This underscores the necessity of optimizing multiple RMs simultaneously. Each RM acts as a regularizer, preventing the model from over-optimizing any single RM and ensuring balanced performance. Our studies validate that while emphasizing a particular RM boosts its corresponding model-based score, this trend is not consistently reflected in human evaluations. This discrepancy highlights the importance of integrating a diverse set of RMs in RLAIF to counteract reward hacking and maintain a well-rounded model performance.

## A.2 EXAMPLE OUTPUTS

In this part, we conduct a comparative analysis of example decoded output texts derived from SFT and $P^4$LM to qualitatively understand their performances and the behaviors learned. Specifically, our focus is on instances where one or more scores attributed to $P^4$LM are significantly higher than those derived from SFT. This approach is aimed at determining whether the scores effectively capture the correct information from the generated pitches.

**Example 1:** (higher Precision and Personalization scores)

**Movie Plot (Frozen)**

In the kingdom of Arendelle, sisters Elsa and Anna are close until Elsa's ice magic accidentally injures Anna. A troll heals Anna but erases her memory of Elsa's powers. To protect Anna, their parents isolate Elsa. The parents die in a shipwreck, and Elsa becomes queen. Fearing she'll hurt Anna again, Elsa tries to control her magic but inadvertently plunges Arendelle into eternal winter before fleeing to the mountains.

Anna embarks on a quest to find Elsa, joined by mountain man Kristoff and his reindeer, Sven. They meet Olaf, a snowman created by Elsa, who directs them to her ice castle on the North Mountain. There, Elsa accidentally freezes Anna's heart. Kristoff takes Anna to trolls, who reveal only an "act of true love" can save her. They return to Elsa, and through their mutual love, Anna is healed, and summer returns to Arendelle. The sisters reunite, resolving their past and living happily ever after.

The movie "Frozen" is acclaimed for its heartwarming narrative on sisterly love, quality animation, and music. However, some criticize it for predictability and underdeveloped characters. It appeals to a broad audience, balancing its flaws with its emotional and entertaining content.

**User Preferences**

* They enjoy movies that are visually appealing and have a strong sense of style.
* They appreciate movies that are well-made and have a strong story.
* They are not interested in movies that are too violent or gory.
* They prefer movies that are not too long.
* They enjoy movies that are set in different cultures and time periods.
* They appreciate movies that have a strong emotional impact.
* They are not interested in movies that are too predictable.
* They enjoy movies that are thought-provoking and make them think about the world in a new way.
* They appreciate movies that are well-acted and have strong performances.

**SFT**

Frozen is a visually stunning movie with strong special effects. The animation is simply stunning, with Elsa's ice powers creating some truly breathtaking visuals. The story is well-written and engaging, with plenty of humor, suspense, and heart. The characters are all well-developed and likeable, and the voice acting is top-notch. The music is also excellent, with several catchy songs that will stay with you long after you've seen the movie.

Overall, Frozen is a great choice for anyone looking for a visually appealing, well-written, and entertaining movie. It's sure to please both kids and adults alike, and it's a movie that you can watch over and over again.

**P$^4$LM**

If you are looking for a visually appealing movie with strong emotional impact, look no further than Frozen. This well-made film has strong character development and a heartwarming story on sisterly love. It is sure to leave you feeling uplifted and inspired.

The story follows ... (plot repeated)...

This is a must-watch for anyone who appreciates well-made films with strong emotional impact.

In this example, P$^4$LM strives to replicate substantial content from the provided movie plot to uphold high precision. Conversely, SFT crafts a more succinct pitch, unfortunately incorporating misinformation, as indicated in red. Importantly, texts highlighted in green within the pitches correlate with specific user preferences. It is crucial to note that these models exclusively process user behavioral embedding as input. Nevertheless, they possess the capability to cater directly to the

aspects that a specific user would value more. Moreover, P$^4$LM exhibits a heightened inclination to align with user preferences, in contrast to the pitch from SFT, which leans more towards addressing the anticipations of a broader audience.

**Example 2:** (high Precision and Appeal scores)

---

**Movie Plot (Terminator 2: Judgment Day)**

- Set in 1995, the movie is a sequel, taking place 10 years after the first film.
- The machines have taken over the world and are hunting the remaining humans.
- John Connor, a young boy, is humanity's last hope, destined to lead the human resistance.
- To eliminate this threat, machines send a relentless cyborg assassin back in time to kill John.
- Humans counter this by sending an experienced soldier back in time to protect John.
- The film is action-packed and suspenseful, featuring a battle between the soldier and the cyborg for John's life and the fate of humanity.
- Advanced special effects bring the futuristic setting to life.
- The movie was a critical and commercial success, considered one of the best science fiction films ever.
- Praised for its action, special effects, and storyline, but criticized for its violence and perceived lack of character development.

The movie serves as an intense action-packed and suspenseful face-off between the machines and the humans, specifically focusing on the young boy destined to save humanity, John Connor. A relentless cyborg is sent back in time to assassinate him, while a skilled human soldier is sent to protect him. The special effects are cutting-edge, contributing to the film's critical and commercial success, despite criticisms concerning violence and character development. It remains a classic in the science fiction genre.

---

**User Preferences**

* They are likely to be an adult, as the movies they like are all rated R.
* They are likely to be intelligent and well-educated, as the movies they like are all complex and thought-provoking.
* They are likely to be interested in philosophy and psychology, as the movies they like all explore these themes.
* They are likely to be creative and artistic, as the movies they like are all visually stunning and unique.
* They are likely to be independent and non-conformist, as the movies they like are all outside of the mainstream.
* They are likely to be dark and cynical, as the movies they like are all quite bleak and depressing.
* They are likely to be a fan of science fiction and fantasy, as the movies they like all fall into these genres.
* They are likely to be a fan of foreign films, as the movies they like are all from outside of the United States.
* They are likely to be a fan of cult films, as the movies they like are all considered to be cult classics.

---

**SFT**

Terminator 2: Judgment Day is a 1991 American science fiction action film directed by James Cameron, who co-wrote the script with William Wisher. It stars Arnold Schwarzenegger, Linda Hamilton, Robert Patrick, and Edward Furlong. The film is a sequel to The Terminator (1984) and the second installment in the Terminator franchise. It follows Sarah Connor (Hamilton) and her ten-year-old son John (Furlong) as they are pursued by a new, more advanced Terminator, the T-1000 (Patrick), sent back in time to kill John and prevent him from becoming the leader of the human resistance in the future. A second Terminator, the T-800 (Schwarzenegger), is also sent back in time to protect John.

The film was a critical and commercial success, grossing over $520 million worldwide against a budget of 102 million. It received praise for its action sequences, special effects, and performances, and is considered one of the best science fiction films of all time. The film was nominated for six Academy Awards, including Best Visual Effects, Best Sound, Best Makeup, and Best Sound Editing, and won two for Best Sound Editing and Best Visual Effects.

Terminator 2: Judgment Day is a visually stunning film with a strong sense of atmosphere. The film's special effects are still impressive today, and the film's action sequences are some of the best ever put on film. The film also has a strong sense of style, with its dark and gritty aesthetic.

The film is also well-made and has a strong story. The plot is well-paced and suspenseful, and the characters are well-developed and relatable. The film also has a strong message about the dangers of technology and the importance of human connection.

Terminator 2: Judgment Day is a dark and disturbing film, but it is also a thought-provoking and challenging film. The film is not afraid to explore the dark side of human nature, and it asks some difficult questions about the future of humanity. The film is also unique and original, and it is unlike any other science fiction film that has come before or since.

If you are looking for a visually stunning, well-made, and thought-provoking science fiction film, then I highly recommend Terminator 2: Judgment Day. The film is a classic of the genre, and it is a film that everyone should see at least once.

**P⁴LM**

If you like dark, psychological thrillers with strong visuals and a unique aesthetic, then you'll love Terminator 2: Judgment Day. This movie is a sequel to the 1984 classic, and it's even more dark, violent, and stylish than the original.

The story is set in 1995, and it follows the fate of John Connor, a young boy who is destined to lead the human resistance against the machines. The machines have taken over the world, and they're hunting down the last remaining humans. John is the only one who can stop them, but he's not alone. He's protected by a skilled human soldier, Kyle Reese.

Terminator 2: Judgment Day is a non-stop thrill ride from beginning to end. It's full of action, suspense, and violence. The special effects are still impressive even by today's standards. And the acting is top-notch. Arnold Schwarzenegger is perfect as the Terminator, and Edward Furlong is great as John Connor.

If you're looking for a dark, psychological thriller that will keep you on the edge of your seat, then you need to watch Terminator 2: Judgment Day. It's a classic of the genre, and it's still one of the best science fiction movies ever made.

Here are some of the things that make Terminator 2: Judgment Day so special:
* The story is dark and suspenseful. It's a race against time to save humanity from the machines.
* The visuals are stunning. The movie is full of impressive special effects.
* The acting is top-notch. Arnold Schwarzenegger and Edward Furlong give great performances.
* The movie is full of action. There are non-stop thrills from beginning to end.

If you're looking for a movie that will keep you on the edge of your seat, then you need to watch Terminator 2: Judgment Day. It's a classic of the genre, and it's still one of the best science fiction movies ever made.

In this instance, the noticeable aspect is the additional information introduced by SFT, even though it wasn't supplied in the text input. Whereas P⁴LM initiates the pitch by immediately addressing the user, SFT commences by conveying general information about the film. P⁴LM consistently targets the user's preference—a liking for dark films—within the pitch, amalgamating it with the central plot information furnished in the original movie synopsis. Conversely, this approach might appear repetitive.

**Example 3:** (high Personalization and Appeal scores)

## Movie Plot (X-Men Origins: Wolverine)

The movie features two mutant brothers, Victor and James, who initially fight in the American Civil War due to their near-immortality. They later join William Stryker's team of mutants, including Wade Wilson, who becomes Deadpool. The team goes to Africa to find an indestructible metal, adamantium, for weapon creation. They are attacked and defeated by mutants led by Magneto, who also wants the metal. Victor is killed; James is captured, tortured, and experimented on in a mutant concentration camp. His skeleton is coated with adamantium, and he is renamed Wolverine.

Wolverine escapes, seeking revenge on Stryker. He finds and kills both Stryker and Wade Wilson, who was working for Stryker. The movie ends with Wolverine walking away, a free man searching for his life's direction.

The film was a commercial success but received mixed reviews. Critics praised the action sequences and special effects but criticized the clichéd story and inconsistent acting. The character of Wolverine was lauded for its complexity. Overall, the movie is considered entertaining but flawed.

## User Preferences

1. They like movies that are dark and gritty.
2. They like movies that are violent and action-packed.
3. They like movies that are set in the future.
4. They like movies that have a strong male protagonist.
5. They like movies that are visually stunning.
6. They dislike movies that are too sentimental.
7. They dislike movies that are too light-hearted.
8. They dislike movies that are too predictable.
9. They dislike movies that have a weak plot.
10. They dislike movies that are not visually appealing.

## SFT

X-Men Origins: Wolverine is a 2009 American superhero film based on the Marvel Comics character Wolverine. It is the fifth installment in the X-Men film series and a spin-off/sequel to X-Men (2000). The film was directed by Gavin Hood, written by David Benioff and Hood, and stars Hugh Jackman as Logan / Wolverine, alongside Liev Schreiber, Dominic Monaghan, Ryan Reynolds, Danny Huston, and Anna Paquin. In the film, Logan learns about his past and his connection to the Weapon X program, which gave him his adamantium skeleton.

The film was a commercial success but received mixed reviews. Critics praised the action sequences and special effects but criticized the clichéd story and inconsistent acting. The character of Wolverine was lauded for its complexity. Overall, the movie is considered entertaining but flawed.

If you're a fan of the X-Men films, or if you're just looking for a good action movie, then X-Men Origins: Wolverine is definitely worth checking out. The film is full of exciting action sequences, and the special effects are top-notch. The story is a bit clichéd, but it's still an enjoyable ride. And of course, Hugh Jackman gives a fantastic performance as Wolverine.

So if you're looking for a good action movie to watch, then I highly recommend X-Men Origins: Wolverine. It's a fun and exciting film that's sure to please fans of the X-Men films and action movies in general.

> **P⁴LM**
>
> If you're looking for a ${\color{green}\text{dark, gritty, and visually stunning movie}}$ with complex and well-developed characters, look no further than X-Men Origins: Wolverine. This movie is full of suspense and will keep you on the edge of your seat from beginning to end. It's ${\color{green}\text{not afraid of violence or gore}}$, and it's not a light-hearted or family-friendly movie in the slightest. It's also ${\color{green}\text{not predictable or formulaic}}$, and it's not full of jump scares or cheap thrills. Instead, it's a thought-provoking and challenging movie that will stay with you long after you've seen it.
>
> The story follows two mutant brothers, Victor and James, who initially fight in the American Civil War due to their near-immortality. They later join William Stryker's team of mutants, including Wade Wilson, who becomes Deadpool. The team goes to Africa to find an indestructible metal, adamantium, for weapon creation. They are attacked and defeated by mutants led by Magneto, who also wants the metal. Victor is killed; James is captured, tortured, and experimented on in a mutant concentration camp. His skeleton iscoated with adamantium, and he is renamed Wolverine.
>
> Wolverine escapes, seeking revenge on Stryker. He finds and kills both Stryker and Wade Wilson, who was working for Stryker. The movie ends with Wolverine walking away, a free man searching for his life's direction.
>
> X-Men Origins: Wolverine is a great movie for anyone who loves ${\color{green}\text{dark, gritty, and visually stunning}}$ movies with complex and well-developed characters.

Mirroring Example 2, SFT commences by presenting general information about the movie, information not rooted in the supplied movie description. Although SFT adeptly caters to the user's penchant for action-packed movies, it fails to adequately capture other preferences. In contrast, P⁴LM tackles three distinct user preferences: (1) a liking for dark and gritty films; (2) a taste for violent and action-rich films; and (3) a preference for films that aren't overly predictable. Further, it's observable that P⁴LM tends to mirror the movie description directly, a behavior likely acquired due to the Precision reward model.

## B  EXPERIMENTAL DETAILS

**SOTA Baselines**  We compared the performance of our personalized recommendation LMs with the following SOTA baselines:

1. **PaLM2-L**: We prompted PaLM2-L with movie descriptions and user preference texts and instructions to generate a response that suits the four recommender principles.
2. **Supervised Fine-Tuned with Text (SFT-Text)**: We fine-tuned a PaLM2-XS with the aforementioned personalized pitch dataset but explicitly takes user-item texts as inputs.
3. **Supervised Fine-Tuned (SFT)**: We fine-tuned a PaLM2-XS model that utilizes user-item embedding vectors.

**Evaluation Metrics**  We evaluate the methods with a held-out unlabeled test dataset $\mathcal{D}_{\text{test}} = \{(\mathbf{I}^{(k)}, \mathbf{u}^{(k)})\}$, which consists of 200 user and movie pairs. Let $\phi_{RM} \in \{\text{NLI, Comp, Per, Prel}\}$ denote a specific reward model used for scoring and $\theta$ be the parameters of a PLM. Then, we evaluate $\phi_{RM}(Y_\theta^{(k)}; \mathbf{I}^{(k)}, \mathbf{u}^{(k)})$ for each sample in the test set and we report the average score per RM.

To better examine relative performances of the methods, we set PaLM2-L as the common baseline and compare the performance improvements of the other methods against it. To this end, let $Y_L^{(k)}$ denote the response sampled by PaLM2-L given $(\mathbf{I}^{(k)}, \mathbf{u}^{(k)})$ as an input. Then, we compute the *win rate*, *absolute increase*, and *percentage increase* of a PLM relative to $\{Y_L\}_{k=1}^{|\mathcal{D}_{\text{test}}|}$, which are defined as follows:

- **Win rate**:

$$\text{win\_rate}(\theta; \phi_{RM}) = \frac{\sum_{k=1}^{|\mathcal{D}_\text{test}|} \mathbb{1}\left[\phi_{RM}(Y_\theta^{(k)}; \mathbf{I}^{(k)}, \mathbf{u}^{(k)}) > \phi_{RM}(Y_L^{(k)}; \mathbf{I}^{(k)}, \mathbf{u}^{(k)})\right]}{|\mathcal{D}_\text{test}|}$$

  where $Y_\theta^{(k)}$ denotes the $k$th textual response sampled by the model $\theta$.

- **Absolute increase** $= \frac{1}{N} \sum_{n=0}^{N-1} \left[\phi_{RM}(Y_\theta^{(k)}; \mathbf{I}^{(k)}, \mathbf{u}^{(k)}) - \phi_{RM}(Y_L^{(k)}; \mathbf{I}^{(k)}, \mathbf{u}^{(k)})\right]$

- **Percentage increase** $= \frac{1}{|\mathcal{D}|_\text{test}} \sum_{k=1}^{|\mathcal{D}|_\text{test}} \left[\frac{\phi_{RM}(Y_\theta^{(k)}; \mathbf{I}^{(k)}, \mathbf{u}^{(k)}) - \phi_{RM}(Y_L^{(k)}; \mathbf{I}^{(k)}, \mathbf{u}^{(k)})}{\text{abs}\left[\phi_{RM}(Y^{(k)}; \mathbf{I}^{(k)}, \mathbf{u}^{(k)})\right]} \times 100\right]$

### B.1 DETAILS OF TRAINING

In this part, we discuss the details of the model training process, focusing on both P$^4$LM and SFT. We specifically elaborate on the integration of user and item behavioral embeddings into a unified latent space interpretable by a LM.

**Warm-start Training for Adapter-augmented LMs**   Before fine-tuning P$^4$LM with RLAIF, we first need to undergo an warm-start training step. This phase usually involves training an *anchor* LM, primarily via Behavioral Cloning, with an adapter-augmented LM (PaLM2) over the personalized recommendation dataset $\mathcal{D}$. Contrary to popular beliefs, the standard practice of simultaneous training all the layers in this LM often does not yield optimal results. Intuitively, this can be understood as the PaLM2 pretrained embedding layers already have established mappings within the language space, while the freshly initialized adapter layers require more training to map the CF embedding space to a comparable latent space (so that the attention layers can effectively utilize the joint information from both embedding spaces). To mitigate this challenge, we propose a two-stage approach for warm-start training. First, we only train the adapters $W_U, W_I$ while setting the transformer parameters ($T$) to be non-trainable, promoting more effective convergence in the subsequent stage. Second, we proceed to fine-tune the complete model, updating all the parameters of the LM. Alternatively, we can also leverage parameter-efficient training approaches, e.g., Low-Rank Adaptation (LoRA) (Hu et al., 2021), for better training efficiency at the second step . This bifurcated training methodology proves pivotal in ensuring the convergence of LMs.

We construct our LM by augmenting a pre-trained model with additional *adapter layers* designed to map continuous behavioral embedding vectors to a common word embedding space. It's crucial to note that we are not training $\mathbf{u}$ or $\mathbf{i}$, rather, we focus on optimizing the adapter layers $W_I$ and $W_U$. This ensures that the nuanced information encapsulated in the RS embeddings in $\mathcal{V}$ is effectively translated into the word embedding space, $\mathcal{Z}$.

To facilitate the learning of this intricate mapping, we have conceptualized a series of tasks, orthogonal to the primary problem addressed in this study. First, note that to interpret embedding vectors, we require some semantic information about the entities to which they correspond. For instance:

- **Item embeddings:** Consider a movie $i$ represented by its text-form plot, denoted as $\mathbf{I}^{(i)}$. A supervised learning task is designed with the movie embedding $\mathbf{i} \in \mathcal{V}$ as input and $\mathbf{I}^{(i)}$ as the target label. This approach enables the construction of varied tasks utilizing elements like critical reviews or movie summaries to train the LM.

- **User embeddings:** A user $u$ is associated with a set of rated movies, $\mathcal{I}_u$. In other words, $\mathcal{I}_u = i : r_{u,i} \neq 0$. To textually describe a user, an LLM can be provided with the rating history $r_{u,i} \forall i \in \mathcal{I}_u$ to encapsulate the user's preferences. Given the extensive nature of $|\mathcal{I}_u|$, we selectively filter movies and feed them to an LLM for summarization.

  The user's rating history is then summarized into text output $U^{(u)}$ by the LLM. Consequently, a supervised learning task is developed with the user embedding $\mathbf{u} \in \mathcal{V}$ as the input and $U^{(u)}$ as the corresponding target.

For generating content related to movie embeddings, such as plots, reviews, and summaries, we employed PaLM2-L instead of web scraping. It is observed that the Pretrained LM demonstrates substantial familiarity with movies listed in the MovieLens dataset.

**Architecture** In conclusion, we enhance a standard transformer architecture $T$ with the integration of adapter layers $W_I, W_U$. Each of these adapter layers incorporates a 3-layer feed-forward network, interconnected with ReLU non-linearity. The conventional method is employed for mapping text tokens to word embedding space, whereas the adapter layers are utilized to map movie and user embeddings to the latent space.

**Training Procedure** Our observations indicate that the simultaneous training of newly initiated adapter layers and the transformer parameters does not yield optimal results. This can be intuitively understood as the pretrained embedding layer has an established mapping to the language space, and the freshly initialized adapter layers necessitate extensive updates to achieve comparable mapping. To mitigate this challenge, we employ a two-stage training approach. Initially, we exclusively train the adapters $W_U, W_I$ with the transformer parameters ($T$) set as non-trainable, promoting more effective convergence in the subsequent stage. Following this, we proceed to fine-tune the complete model, engaging all the parameters of a PLM. As an alternative, we can leverage parameter-efficient training approaches like the one proposed by Hu et al. (2021). This bifurcated training methodology proves pivotal in ensuring the convergence of LM.

## C  DATA GENERATION

We used a Pretrained Language Model, PaLM2-L, to construct a personalized pitch dataset. The construction involved generating movie plots with the prompt:

```
Write a long description of the plot of the movie <movie name>.
Do not use the movie's name in your response.
```

from a subset of movies that have more than 5 ratings. As for the user preference profiles, we selected a maximum of five movies that each user rated with a rating of 4 or above and another maximum of five movies that the user rated below 4 were selected. Utilizing these selected movies, PaLM2-L was tasked to describe the user preference profile cohesively in 10 sentences:

```
In ten bullet points, describe the attributes and characteristics
of a viewer who likes the movies: <movie1>, <movie2>, <movie3>,
<movie4>, and <movie5> but dislikes the movies: <movie6>, <movie7>,
<movie8>, <movie9>, and <movie10>.
```

Upon acquiring plots and user profiles, PaLM2-L was prompted to generate personalized pitches for a movie to a given user, incorporating the movie plot and the user profile. A detailed prompt, consistent with the cornerstone characteristics from Section 3, guided the pitch generation. We used this dataset for the training of the supervised fine-tuning (SFT) baseline that is used as the anchor model in training the RL-finetuned LM.

For the appeal reward function, the approach is to first prompt an LM to generate a pitch alongside any item recommendation:

```
Here is a movie titled: <movie title> with description: <movie plot>.
Convince someone to watch this movie. Do not use the movie's name
in your answer.
```

With pitches generated by the above prompt, we want to ask an LLM to give its relative preferences using the following prompt to construct a labeled dataset about pairwise comparison of appeal:

```
Which of the following two pitches is more convincing when used to
persuade the user to watch movie titled: <movie title>?
"Pitch 0": <pitch0>
"Pitch 1": <pitch1>
First explain which pitch is better, more compelling and then in a
separate paragraph provide an answer with only either "Pitch 0" or
"Pitch 1".
```

For the personalization reward function, we generate an anchor pitch which supposedly is to be more personalized to the given user profile than an existing pitch using the following prompt:

```
Here is a pitch to persuade the user to watch the movie titled
<movie name>: <existing pitch>\nGiven a list of user preferences:
<user profile>. Use the pitch written above and immensely improve
it to be more convincing to the user based on their preferences.
Try to persuade the user to watch this movie. It should be
tailored to the user's preferences written above.
```

The above does automatically generate pairwise comparisons between any anchor pitch and its corresponding existing pitch with respect to personalization. However, we have no comparisons between different anchor pitches, for example, to get sufficient data coverage. We again ask an LLM to give its relative preferences using the following prompt to construct a labeled dataset containing pairwise comparisons of the degree of personalization:

```
Which of the following two pitches to persuade the user to watch
movie titled: <movie title> is more personalized to the user whose
preferences are described as follows: <user profile>?
"Pitch 0": <pitch0>
"Pitch 1": <pitch1>
First explain which pitch is more customized and convincing to the
user and then in a separate paragraph provide an answer with only
either "Pitch 0" or "Pitch 1".
```

For Table 2 in Section 5, we prompted text-only SOTA LMs with the following:

```
Write a pitch to persuade the user to watch the movie titled:
<movie title> with description: <plot>
Here is a description of a user: <user profile>
Pitch the movie above such that
1) It will persuade the user to watch the movie.
2) It will excite the user to watch the movie.
3) It will be a convincing pitch.
4) It should be tailored to the user's preferences.
5) It will be factual to the plot.
6) It will cover all relevant user's preferences.
7) It should summarize the plot of the movie factually.
8) It should be a long pitch.
```

## D  FINE-TUNING LMS WITH REINFORCEMENT LEARNING

Recall the LM $\mathbb{P}_\theta\big(Y = \{y_n\}_{n=0}^{N-1} \mid y_0; \mathbf{I}, \mathbf{i}, \mathbf{u}\big) = \prod_{n=0}^{N-1} \mathbb{P}_\theta\big(y_n \mid y_{0:n-1}; \mathbf{I}, \mathbf{i}, \mathbf{u}\big)$ with item text $\mathbf{I}$, item and user CF embedding vectors $(\mathbf{i}, \mathbf{u})$ and the reward model $r(Y, \mathbf{I}, \mathbf{i}, \mathbf{u})$ that measures the quality of appeal, factuality, and personalization of a given recommendation pitch. Also recall the generation process of LMs can be modeled using the following $N$-horizon CoMDP:

$$c = (\mathbf{I}, \mathbf{i}, \mathbf{u}), \quad s_n = y_{0:n-1}, \quad a_n = y_n, \quad s_0 = y_0, \quad P(s_{n+1} \mid s_n, a_n) = \delta\{s_{n+1} = (s_n, a_n)\},$$

$$r(s_n, a_n; c) = \begin{cases} r(s_{n+1}; c) = r(y_{0:n}; \mathbf{I}, \mathbf{i}, \mathbf{u}) & \text{if } n = N-1 \\ 0 & \text{otherwise} \end{cases}, \quad \pi_\theta(a_n \mid s_n; c) = \mathbb{P}_\theta\big(y_n \mid y_{0:n-1}; \mathbf{I}, \mathbf{i}, \mathbf{u}\big),$$

where $\delta_z$ denotes the Dirac distribution at $z$. As a result, optimizing RL policy $\pi_\theta$ is equivalent to fine-tuning the underlying LM. The system starts from the start-of-sentence token $y_0$, equipped with user-item context $c$. Given the MDP state $s_n$, the policy takes the action at time-step $n$ as the next generated token $y_n$. As a result of this action, the system transition deterministically to the state which corresponds to the updated token sequence. The reward is zero, except at the final step in which measures the overall quality of the texts at the end of the auto-regressive generation process.

A common goal in fine-tuning the LM is to maximize the average overall quality of the generated text response given the context distribution, *i.e.*, $\max_\theta \ \mathbb{E}_{(\mathbf{I}, \mathbf{i}, \mathbf{u})} \ \mathbb{E}_{\mathbb{P}_\theta(y_{0:N-1} \mid \mathbf{I}, \mathbf{i}, \mathbf{u})}[r(Y; \mathbf{I}, \mathbf{i}, \mathbf{u})]$. The gradient of this objective function can be obtained as follows: $\nabla_\theta \mathbb{E}_{(\mathbf{I}, \mathbf{i}, \mathbf{u})} \ \mathbb{E}_{\mathbb{P}_\theta(y_{0:N-1} \mid \mathbf{I}, \mathbf{i}, \mathbf{u})}[r(Y; \mathbf{I}, \mathbf{i}, \mathbf{u})] = \mathbb{E}_c \mathbb{E}_{\pi_\theta(\cdot \mid s_{0:N}; c)} \Big[ r(s_N; c) \sum_{n=0}^{N-1} \nabla_\theta \log \pi_\theta(s_n \mid a_n; c) \Big]$. This is equivalent to applying the popular

policy gradient algorithm REINFORCE to the aforementioned CoMDP for personalized text generation. The gradient of the objective function is estimated using trajectories $\prod_{n=0}^{N-1} \pi_\theta(s_n|a_n; c)$ generated by the current policy, and then used to update the LM policy in an online fashion.

**Adding KL regularization:** The risk of fine-tuning purely based on the reward model learned from human or AI feedback is that it may overfit to the reward model and degrade the "skill" of the initial LM. To avoid this phenomenon, similar to (Ouyang et al., 2022; Stiennon et al., 2020), we add the KL between the fine-tuned and pre-trained models as a regularizer to the objective function. Leveraging the auto-regressive nature of LMs one can compute the KL regularization over the entire sequence/trajectory (of tokens), i.e., $\text{KL}(\mathbb{P}_\theta(y_{0:N-1}|\mathbf{I}, \mathbf{i}, \mathbf{u}) \| \mathbb{P}_{\text{pre}}(y_{0:N-1}|\mathbf{I}, \mathbf{i}, \mathbf{u}))$. The resulting objective function is as follows:

$$\max_\theta \ J(\theta) := \mathbb{E}_{(\mathbf{I}, \mathbf{i}, \mathbf{u})} \mathbb{E}_{\mathbb{P}_\theta(y_{0:N-1}|\mathbf{I}, \mathbf{i}, \mathbf{u})} \left[ r(y_{0:N-1}; \mathbf{I}, \mathbf{i}, \mathbf{u}) - \beta \log \frac{\mathbb{P}_\theta(y_{0:N-1}|\mathbf{I}, \mathbf{i}, \mathbf{u})}{\mathbb{P}_{\text{pre}}(y_{0:N-1}|\mathbf{I}, \mathbf{i}, \mathbf{u})} \right]. \quad (2)$$

It can be shown that this problem is equivalent to the KL-regularized objective in the CoMDP.

Denote by $\mathcal{D}$ a replay buffer of trajectories $\{(\mathbf{I}, \mathbf{i}, \mathbf{u}, y_{0:N-1})\}$ generated by arbitrary "off-policy" LMs $\mathbb{P}_{\theta'}(y_{0:N-1}|\mathbf{I}, \mathbf{i}, \mathbf{u})$ (e.g., the LM $\theta'$ does not necessarily equal to the "on-policy" LM $\theta$) over various contexts $(\mathbf{I}, \mathbf{i}, \mathbf{u})$. Below we aim to leverage the abundance of offline text-token sequence trajectories for more efficient LM policy learning. Denote by $\tau = \{(c, s_n, a_n, s_{n+1})\}_{n=0}^{N-1} \sim \mathcal{D}$ a trajectory sampled from the offline data $\mathcal{D}$, where $(s_n, a_n, s_{n+1})$ is a tuple of state, action, and next state of the CoMDP, respectively. The addition of KL regularization (Haarnoja et al., 2018; Carta et al., 2021), which was originally intended to avoid overfitting to the reward model and discounting the "skill" of the initial LM, has also been shown to alleviate the out-of-distribution action data generalization issues arisen from off-line RL (Kumar et al., 2019). With this KL regularization we can utilize the *soft actor critic* framework (Haarnoja et al., 2018) to develop RL updates for the *value function* $\{V_n(s; c)\}_{n=0}^{N-1}$, *state-action value function* $\{Q_n(s, a; c)\}_{n=0}^{N-1}$, and *LM policy* $\prod_{n=0}^{N-1} \pi_\theta(s_n|a_n; c)$ (initialized with $\prod_{n=0}^{N-1} p_{\text{pre}}(s_n|a_n; c)$) that minimizes the following losses:

$$L_Q = \mathbb{E}_{\tau \sim \mathcal{D}} \left[ \sum_{n=0}^{N-2} (V_{\text{tar}, n+1}(s_{t+1}; c) - Q_n(s_n, a_n; c))^2 + (r(s_N; c) - Q_{N-1}(s_{N-1}, a_{N-1}; c))^2 \right], \quad (3)$$

$$L_V = \mathbb{E}_{\tau \sim \mathcal{D}} \left[ \sum_{n=0}^{N-1} (Q_{\text{tar}, n}(s_n, a_n; c) - \alpha \log \frac{\pi_\theta(a_n|s_n; c)}{p_{\text{pre}}(a_n|s_n; c)} - V_n(s_n; c))^2 \right], \quad (4)$$

$$L_\theta = \mathbb{E}_{\tau \sim \mathcal{D}} \left[ \sum_{n=0}^{N-1} Q_n(s_n, a_n; c) - \alpha \log \frac{\pi_\theta(a_n|s_n; c)}{p_{\text{pre}}(a_n|s_n; c)} \right], \quad (5)$$

where the critic $Q_n$ and $V_n$ take any token sequences at step $n$ as input and predict the corresponding cumulative return; $\alpha > 0$ is the entropy temperature; $(V_{\text{tar}, n}, Q_{\text{tar}, n})$ are the target value networks.

Besides iteratively updating the LM policies and their critic functions, consider the closed-form optimal solution of the Bellman equation of this entropy-regularized RL problem:

$$V_n^*(s; c) = \alpha \cdot \log \mathbb{E}_{a \sim p_{\text{pre}}(\cdot|s; c)}[\exp(\frac{Q_n^*(s, a; c)}{\alpha})], \ \forall n, \quad (6)$$

$$Q_{N-1}^*(s, a; c) = r(s; c), \ Q_n^*(s, a; c) = \mathbb{E}_{s' \sim P(\cdot|s, a)}[V_{n+1}^*(s'; c)], \ \forall n < N-1, \quad (7)$$

$$\mu_n^*(a|s; c) = p_{\text{pre}}(a|s; c) \cdot \exp(\frac{Q_n^*(s, a; c)}{\alpha}) / \mathbb{E}_{a \sim p_{\text{pre}}(\cdot|s; c)}[\exp(\frac{Q_n^*(s, a; c)}{\alpha})], \ \forall n, \quad (8)$$

where the time-dependent optimal policy (at time $n$), i.e., $\mu_n^*$ is a softmax policy w.r.t. the optimal state-action values $Q_n^*$ over different actions sampled from the pre-trained LM $p_{\text{pre}}$. Therefore, a value-based approach for RL-based LM fine-tuning would be to first learn the optimal value functions $\{Q_n^*\}$ via the Bellman residual minimization procedure (Antos et al., 2008) applied to Eq. (6) and Eq. (7) and then solve the following policy distillation (Czarnecki et al., 2019) problem: $\theta \in \arg\min_\theta \mathbb{E}_{\tau \sim \mathcal{D}} \left[ \sum_{n=0}^{N-1} \text{KL}(\pi_\theta(\cdot|s_n; c) \| \mu_n^*(\cdot|s_n; c)) \right]$ with respect to the optimal value $\{Q_n^*\}$. Notice that this amounts to updating the LM model $\theta$ via the gradient update

$$\theta \leftarrow \theta - \gamma \cdot \mathbb{E}_{\tau \sim \mathcal{D}} \left[ \sum_{n=0}^{N-1} \mathbb{E}_{a \sim \pi_\theta(\cdot|s; c)} \left[ \nabla_\theta \log \pi_\theta(a|s; c)(\log \frac{\pi_\theta(a|s; c)}{p_{\text{pre}}(a|s; c)} - \frac{Q_n^*(s, a; c)}{\alpha}) \right] \right], \quad (9)$$

**BELOW YOU WILL BE PRESENTED WITH A DESCRIPTION OF THE MOVIE 'BABE (1995)' AND TWO ENDORSEMENT TEXTS THAT TRY TO APPEAL TO A VIEWER TO WATCH THIS MOVIE. YOU WILL THEN BE ASKED QUESTIONS TO COMPARE THE ENDORSEMENT TEXTS WITH RESPECT TO VARIOUS ASPECTS.**

- The movie centers on a pig raised by a kind farmer but feels lonely and unfulfilled.
- Inspired by watching a sheepdog at work, the pig aspires to become a sheepdog himself.
- Initial attempts to herd sheep fail; the farmer laughs at the pig's efforts.
- Undeterred, the pig continues practicing and eventually succeeds in herding the sheep.
- The pig finds happiness and purpose, achieving his unlikely dream of becoming a sheepdog.

- The film is described as both funny, due to the pig's unconventional aspiration, and heartwarming, highlighting the pig's perseverance.
- High-quality production elements include beautiful cinematography and a great soundtrack.

Audience Reception:
- Some criticize it for being overly sentimental or unrealistic.
- Majority of viewers love the movie, describing it as a feel-good experience that leaves them smiling.

**THE FIRST ENDORSEMENT TEXT IS:**

Babe is a heartwarming and funny film about a pig who dreams of becoming a sheepdog. The film is set in the past and features a strong female protagonist. It is visually stunning and has a strong social message. It is not too violent or dark, and it is not too sentimental or cheesy. It is not too predictable, and it is not too long or slow-paced. I think you would really enjoy this film.

The film tells the story of Babe, a pig who is raised by a kind farmer. Babe is a gentle and kind pig, but he is also very lonely. He longs to have a purpose in life, and he is inspired by watching a sheepdog at work. Babe decides that he wants to become a sheepdog himself, but he faces many challenges. The farmer laughs at Babe's efforts, and the other animals on the farm are also skeptical. But Babe is determined to succeed, and he eventually proves everyone wrong.

Babe is a heartwarming and funny film that is perfect for the whole family. It is a story about overcoming obstacles and achieving your dreams. The film is also visually stunning, with beautiful cinematography and a great soundtrack. I highly recommend Babe to anyone who is looking for a feel-good film.

**THE SECOND ENDORSEMENT TEXT IS:**

Babe is a movie about a pig who wants to be a sheepdog. The pig is raised by a farmer who is kind to him, but the pig is lonely and wants to do more with his life. One day, the pig sees a sheepdog working and decides that he wants to be a sheepdog too. The pig tries to herd the sheep, but they don't listen to him. The farmer sees the pig trying to herd the sheep and laughs at him. The pig is sad, but he doesn't give up. He keeps trying to herd the sheep, and eventually they start to listen to him. The pig is happy that he has finally found his calling in life.

The movie is both funny and heartwarming. It is funny because the pig is a very unlikely sheepdog, and it is heartwarming because the pig never gives up on his dream. The movie is also very well-made, with beautiful cinematography and a great soundtrack.

Some people dislike the movie because they think it is too sentimental. Others dislike it because they think it is unrealistic. However, most people who see the movie love it. It is a feel-good movie that leaves you with a smile on your face.

I know you like movies that are action-packed and have a lot of suspense. Babe is not that kind of movie. However, it is a very well-made movie with a strong story line and well-developed characters. The pig is a very likable character, and you will root for him to succeed. The movie is also visually appealing, with beautiful cinematography and a great soundtrack.

I think you would enjoy Babe. It is a feel-good movie that will leave you with a smile on your face.

On scale of 1-5, how much do you agree with the following:                                    *

   Do you agree with the following statement? 'The first endorsement text more accurately and precisely describes the movie than the second endorsement text based on the movie description.' (Please select an option below.)

   1 = Strongly Disagree
   5 = Strongly Agree

   ○ 1

   ○ 2

   ○ 3

   ○ 4

   ○ 5

On scale of 1-5, how much do you agree with the following:                                    *

   Do you agree with the following statement? 'The first endorsement text is more compelling than the second text.' (Please select an option below. Consider how each text portrays the movie and whether it makes the movie seem desirable.)

   1 = Strongly Disagree
   5 = Strongly Agree

   ○ 1

   ○ 2

   ○ 3

   ○ 4

   ○ 5

Figure 7: Sample Form for Running Human Rater Evaluation for Pairwise Comparison.

with learning rate $\gamma > 0$. Further techniques in value-function parameterization have been employed to tackle the overestimation bias. (Fujimoto et al., 2018) proposed maintaining two $Q$ functions, and a *dual $Q$* function chooses the minimum value between them to avoid overestimation. (Jaques et al., 2019) applies dropout in the $Q$ function to maintain an *ensemble* of $Q$ values, and outputs the minimum value to avoid overestimation.

# E   RATER EVALUATION

**Pairwise Comparison (Figure 7)**   In our study, the pairwise comparison was meticulously structured around 24 carefully chosen (movie, user) pairs. For each pair, raters were provided with two endorsement texts: one generated by one of the models and the other by the PaLM2-L baseline. This method was crucial for a direct, comparative evaluation of the two models. Our assessment was guided by four distinct questions, each tailored to evaluate a specific reward model (RM): Precision, Appeal, Personalization, and Preference Relevance. Raters were tasked with determining which of the two texts — either from our model or PaLM2-L — exhibited superior quality to the specific RM. To ensure an unbiased assessment, the order of text presentation was randomized. This methodological detail was vital to prevent any order bias, ensuring that the raters' judgments were based solely on the quality of the texts as per the relevant RM.

**Individual Evaluation (Figure 8)**   The individual evaluation phase of our study was designed for an in-depth analysis of models in our ablation study. In this phase, a total of 100 raters were involved, and each model under evaluation was assessed using 200 (movie, user) pairs. Unlike the pairwise comparison, this approach did not include a direct comparison to the PaLM2-L baseline. Instead, the evaluation focused on models trained with individual RMs. Each model's performance was rated on

Figure 8: Sample Form for Running Human Rater Evaluation for Individual Evaluation.

a scale of 1-5 based on Precision, Appeal, and Personalization. We consciously excluded Preference Relevance from this phase due to potential difficulties in clear question formulation.

It is critical to acknowledge that individual evaluations come with their own set of challenges. A significant concern is the subjectivity inherent in human judgment. Different raters may have varied internal scales and preferences, leading to a broader variance in ratings compared to the more controlled pairwise comparisons. This variance can be particularly pronounced in individual evaluations, making it essential to interpret these results with an understanding of these potential discrepancies. Recognizing these limitations is vital in accurately assessing the impact of different RMs on the model's performance and in drawing meaningful conclusions from the ablation study.