# OpenReview forum: "Factual and Personalized Recommendation Language Modeling with Reinforcement Learning"
_ICLR.cc/2024/Conference — Submitted to ICLR 2024_

### Official Review · Reviewer_SpKz · 2023-10-29

**Soundness:** 2 fair
**Presentation:** 3 good
**Contribution:** 1 poor
**Rating:** 5
**Confidence:** 3

**Summary:**

This paper introduces a P4LM framework that provides recommendations, taking into account the precision, personalization, appeal, and preference relevance of items to users. Reward functions are crafted for each perspective, involving the training of reward models based on AI-labeled data derived from PaLM2-LLM. Subsequently, a joint reward function is formulated, employing reinforcement learning to master the text generation policy.

**Strengths:**

- Overall, the authors present insightful perspectives for evaluating the effectiveness of conversational Recommender Systems (RS).

- The paper articulates the overall process with clarity, making the framework straightforward to implement.

- Including human evaluation, the paper offers intriguing insights into the reward hacking issue encountered during training with a singular reward.

**Weaknesses:**

- While the authors' perspectives on assessing model effectiveness are noteworthy, the rationale behind the proposed framework's efficacy in enhancing precision, personalization, appeal, and user preference relevance remains ambiguous. The reward functions, trained via annotations from an LLM model, may not necessarily echo the authentic experiences of users. Moreover, practical recommendation scenarios are complex, and it is uncertain whether these nuances are effectively captured by a reward model.

- The experiments conducted primarily assess the model concerning the targeted reward functions, showing that incorporating RL improves performance on these specific metrics—a finding that is somewhat predictable. It would be advisable to include human evaluations when assessing baseline methods to more convincingly demonstrate practical effectiveness.

- The paper's technical contribution seems limited and under-assessed. The crux of the proposed method's novelty resides in the design of LLM-based reward functions. However, the validation of these reward functions is lacking, and no clear insights are given on how these rewards enhance recommendations. This omission leaves the paper's contributions indistinct.

- The experimental validations are not comprehensive. The reliance on a single dataset, along with a comparison with only three baselines—two of which are merely variants, and the other, the foundational model also used for data generation. There is a need for broader evaluations against additional baselines, datasets, and models to affirm the method's effectiveness.

- In Table 1, the evaluation scores for preference relevance metrics fall below those of other baselines, casting doubt on the assertion of superiorly elucidating project characteristics and their ties with user preferences. This discrepancy warrants an explanation to reconcile the claims with the empirical data.

**Questions:**

Please refer to the weaknesses

---

> ### Author Response · Authors · 2023-11-16
> **Author Response**
>
> __[Key Items]__
> * [Evaluation] We have added auto-evaluation results with data generated by GPT-4. Human evaluations are also being conducted. Please see the common response.
> * [Amazon dataset] We are training our models using the Amazon product dataset, and we will share the results and update the paper as soon as they are available. Please see the common response.
>
> __[Our Detailed Responses]__
>
> Thank you for your detailed review and constructive criticism of our work. We appreciate the opportunity to address each of your concerns and provide further clarity on our approach and findings.
>
> __> The reward functions, trained via annotations from an LLM..__
>
> We acknowledge your concerns about the RMs. To further support their authenticity we have conducted additional auto-evaluations (via GPT-4 generated test sets) and human-rater evaluations are well under way. Please see the common response for more details.
>
> __> advisable to include human evaluations…__
>
> Please check our common response.
>
> __> insights are given on how these rewards enhance recommendations__
>
> Thank you for your insightful question. We would like to clarify that this work specifically addresses the generation of factual, compelling, and personalized endorsement statements (or “pitches”) of a specific item. The broader question of how these endorsement texts impact real users’ engagement and how a conversational recommender should continue the interaction is left for future research.
>
> __> The reliance on a single dataset...__
>
> Please check our common response.
>
> __> In Table 1, … preference relevance metrics fall below those of other baselines…__
>
> Regarding the lower Preference Relevance scores observed in our experiments in Table 1, we have discussed a potential reason in the paper (Page 7): P4LM may prioritize other aspects of endorsement quality at the expense of preference relevance. Additionally, it's important to note that a good endorsement can effectively focus on specific user preferences, even if it doesn't cover the entire spectrum. However, incorporating a Preference Relevance RM can serve as a regularization mechanism, ensuring that endorsements still attempt to encompass a broad range of user preferences. Furthermore, in multi-objective optimization, it's often not feasible to enhance all objectives simultaneously, as illustrated by the Pareto front concept, indicating inevitable trade-offs among optimized rewards. As shown in Table 2, focusing on a single RM can yield high Preference Relevance scores, but at the cost of performance in other RMs. Practically, prioritizing a particular metric as more important allows for the adjustment of reward weights to reflect this preference.
>
> We hope these responses adequately address your concerns and demonstrate our commitment to rigorously validating and improving our work. We look forward to any further feedback you may have.

---

> > ### Author Response · Authors · 2023-11-23
> > **Request to Reviewer SpKz**
> >
> > Dear Reviewer,
> >
> > Thank you so much for your time and efforts in reviewing our paper. We have addressed your comments in detail and are happy to discuss more if there are any additional concerns. We are looking forward to your feedback and would greatly appreciate you consider raising the scores. For an overview of the latest updates and changes we've made, please refer to our common response comment.
> >
> > Thank you,
> >
> > Authors

---

### Official Review · Reviewer_VcLv · 2023-10-31

**Soundness:** 3 good
**Presentation:** 3 good
**Contribution:** 3 good
**Rating:** 6
**Confidence:** 3

**Summary:**

This paper proposes a framework called P4LM that generates personalized narratives of an item. It would be useful to be equipped by a conversational RS to enhance user experience.

The model incorporates the (user and item) embedding spaces of a recommender system, and use RLAIF (RL from AI feedback, the dataset they used to finetune LM or train reward model is generated from prompting a PaLM-L) to fine-tune a language model with reward models including precision, appeal, personalization, and preference relevance.

The method is evaluated on the MovieLens dataset.

**Strengths:**

1. Generating personalized narrative is an interesting and useful problem, and seems unexplored in the literature before.
2. The proposed RLAIF-based framework is general, and could be applied to other recommendation datasets.

**Weaknesses:**

1. There is no human evaluation on comparison among P4LM, P4LM-S, SFT, SFT-Text, PaLM-L. How do we know that P4LM is actually better than others in terms of real human feedbacks? Also, PaLM-L’s samples are not provided in appendix.
2. The authors only experiment on one dataset. I understand the complexity of the whole procedure, but it this paper would be much stronger if the proposed method could be validated another dataset.
3. I didn't search very carefully, but the author didn't compare the proposed method with any baselines from other papers. Or if the problem is completely new (I am not sure), then this would not be an issue.

**Questions:**

1. What recommender system is used for extracting the embeddings? How does recomender system performance affect this task?
2. How does P4LM and SFT take user/item embeddings as input? How does SFT take user and item descriptions as input? What is the detailed network design for these parts?

---

> ### Author Response · Authors · 2023-11-16
> **Author Response**
>
> __[Key Items]__
> * [Human evaluation] Please check our common response. In short, we are conducting extensive human evaluations, and we will include the results as soon as the results become available.
> * [Amazon product dataset] Please check our common response. In short, we are training our models with the Amazon dataset. We plan to share the results as soon as they are available.
>
> __[Our Detailed Responses]__
>
> Thank you for your insightful review and pertinent questions. Below, we address each of your points and questions in detail.
>
> __> Human evaluation__
>
> We acknowledge the importance of human evaluation in assessing the effectiveness of P4LM compared to other models. To this end, we are conducting comprehensive human evaluations comparing P4LM, P4LM-S, SFT, SFT-Text, and PaLM-L. These evaluations will provide crucial insights into real human feedback on our model's performance. We will update the paper with these results as soon as they are available. For more details on this process, please refer to our common response.
>
> __> Another dataset__
>
> We agree with your assessment regarding the need for validating our method on more than one dataset. To address this, we are currently training P4LM on the Amazon product dataset, which will significantly broaden our evaluation scope. The results from this expanded dataset will be included in our paper as soon as they are available, strengthening our findings.
>
> __> Additional baselines__
>
> To the best of our knowledge, our work is the first to tackle the specific problem of generating endorsement texts with defined characteristics in a recommendation setting. Therefore, direct comparisons with existing baselines in other papers may not be straightforward. However, we are open to including additional relevant baselines if suggested.
>
> __[Question 1]__
>
> In our study, we employed matrix factorization-based collaborative filtering to generate user-and-item behavioral embeddings whose dot product predicts the rating of the recommendation. While our problem assumes the embedding vectors are provided as data input and P4LM is general enough to take different forms of embedding vectors as input, we acknowledge that more informative latent embeddings from a more advanced recommender system could potentially enhance the quality of generated endorsement texts. Improving the quality of the behavioral embedding space is left for future work.
>
> __[Question 2]__
>
> Regarding the detailed network design of P4LM and SFT, please see Appendix B.1 of our paper for details. In a nutshell, we employed Adapter layers to transform the user/item behavioral (CF) embeddings into a latent space compatible with the language model's architecture. This technique allows the language model to effectively integrate and utilize user and item embedding information to generate endorsements.
>
> We hope this response addresses your concerns and questions. We are committed to enhancing our work based on your feedback and look forward to any further insights you may have.

---

> > ### Author Response · Authors · 2023-11-23
> > **Request to Reviewer VcLv**
> >
> > Dear Reviewer,
> >
> > Thank you so much for your time and efforts in reviewing our paper. We have addressed your comments in detail and are happy to discuss more if there are any additional concerns. We are looking forward to your feedback and would greatly appreciate you consider raising the scores. For an overview of the latest updates and changes we've made, please refer to our common response comment.
> >
> > Thank you,
> >
> > Authors

---

### Official Review · Reviewer_wWN5 · 2023-11-03

**Soundness:** 3 good
**Presentation:** 3 good
**Contribution:** 2 fair
**Rating:** 5
**Confidence:** 2

**Summary:**

The paper discusses the problem of language modeling for personalized recommendation, which aims to generate natural language responses that explain the relevance of recommended items to users’ preferences. The paper proposes a novel approach called P4LM, which leverages reinforcement learning with AI feedback to fine-tune a pre-trained language model with four reward models that measure precision, appeal, personalization, and preference relevance of the generated responses. The paper demonstrates the effectiveness of P4LM on a conversational movie recommendation task, using the MovieLens 25M dataset. The paper shows that P4LM can generate factual, personalized, and compelling movie endorsements that better align with users’ preferences and item features.

**Strengths:**

Originality: The paper proposes a novel approach to language modeling for personalized recommendation, which leverages reinforcement learning with AI feedback to fine-tune a pre-trained language model with four reward models that measure personalization, precision, appeal, and preference relevance.
Quality: The paper is well-written and organized, with clear problem formulation, methodology, and evaluation. The paper provides sufficient background and related work on recommender systems, language models, and reinforcement learning.
Significance: The paper addresses a challenging and important problem of generating factual, personalized, and compelling recommendation endorsements that better explain item characteristics and their relevance to user preferences. The paper also has practical implications for enhancing the user experience and satisfaction in conversational recommender systems.

**Weaknesses:**

1) The paper does not explain why existing methods are insufficient or what are the specific challenges and opportunities in this domain.
2) The paper provides a comprehensive literature review of related work, especially on conversational recommender systems, language models, and reinforcement learning. But it only cites those papers without discussing their strengths and limitations or comparing them with the proposed approach.
3) Paper does not discuss the assumptions and limitations of the approach.
4) The paper does not describe the implementation details and hyperparameters of the proposed approach, such as the size of the models, and the reinforcement learning algorithms. For example: "where η1, η2, η3, η4 ≥ 0 are importance weights for the component rewards, and are treated as hyper-parameter"
5) The evaluation is not rigorous for an applied paper. One dataset is not enough to draw conclusions.
6) The paper does not present any qualitative analysis or examples of the generated recommendation texts by the proposed approach. It only shows quantitative scores based on model-based metrics, which may not reflect the true quality and diversity of the texts.
7) What are the advantages and disadvantages of using adapter layers to augment the language model?

8) How does the P4LM model deal with cold-start problems, where there is not enough user or item information available?
9) How does the P4LM model compare with other conversational recommender systems that use different language models or architectures?
10) Can authors talk more about  trade-offs between different reward models, such as precision, appeal, personalization, and preference relevance

**Questions:**

1) The paper does not explain why existing methods are insufficient or what are the specific challenges and opportunities in this domain.
2) The paper provides a comprehensive literature review of related work, especially on conversational recommender systems, language models, and reinforcement learning. But it only cites those papers without discussing their strengths and limitations or comparing them with the proposed approach.
3) Paper does not discuss the assumptions and limitations of the approach.
4) The paper does not describe the implementation details and hyperparameters of the proposed approach, such as the size of the models, and the reinforcement learning algorithms. For example: "where η1, η2, η3, η4 ≥ 0 are importance weights for the component rewards, and are treated as hyper-parameter"
5) The evaluation is not rigorous for an applied paper. One dataset is not enough to draw conclusions.
6) The paper does not present any qualitative analysis or examples of the generated recommendation texts by the proposed approach. It only shows quantitative scores based on model-based metrics, which may not reflect the true quality and diversity of the texts.
7) What are the advantages and disadvantages of using adapter layers to augment the language model?

8) How does the P4LM model deal with cold-start problems, where there is not enough user or item information available?
9) How does the P4LM model compare with other conversational recommender systems that use different language models or architectures?
10) Can authors talk more about  trade-offs between different reward models, such as precision, appeal, personalization, and preference relevance

---

> ### Author Response · Authors · 2023-11-16
> **Author Response**
>
> __[Key Items]__
> * [Evaluation] Please see the common response regarding human evaluations and the experiments using the Amazon dataset.
> * [Rewriting] More paper revisions are underway as we are incorporating new results and analysis. Please stay tuned!
>
> __[Our Detailed Responses]__
>
> Thank you for your thorough review and insightful questions. Your feedback is instrumental in helping us refine our work. We address each of your points and questions as follows:
>
> 1. To the best of our knowledge, we are the first to address the problem of generating personalized recommendation endorsement texts, which has significant implications as more online recommenders adopt LLMs to interact with users. The key challenges we address include the absence of datasets for personalized endorsements and the standardized evaluation metrics to quantify the quality of endorsements. Our contributions are: identifying key characteristics that contribute to good endorsements, demonstrating quantification methods for these characteristics, using these metrics to evaluate and RL-finetune LLMs, and substantiating our findings with human evaluations.
>
> 2. We acknowledge that the related work section could be more comprehensive in comparing our approach with existing literature. We will revise this section to include qualitative comparisons and better position our work in the context of existing research.
> 3. We will include a discussion of the assumptions (behavioral embeddings of user representations in recommender systems are given) and limitations (to specific items endorsements) of our approach in the revised version of the paper.
> 4. We will update Appendix B to provide detailed information about the hyperparameters. As for the reinforcement learning algorithm used in our study, we have discussed it in length in the Appendix. We kindly ask the reviewer to check Appendix D.
> 5. We agree that evaluating our approach on a single dataset is limiting. As mentioned in our common response, we are extending our experiments to the Amazon product dataset.
> 6. Generated texts and their qualitative analyses are included in Appendix A of our paper. Additionally, Table 3 presents human evaluation results for the model trained with a single reward. We are conducting further human evaluations to compare P4LM against baselines more rigorously.
> 7. We appreciate your question about the use of adapter layers. These layers are a practical choice that allows us to avoid the introduction of excessive trainable parameters. With the adapter layers, we can incorporate exogenous information such as behavioral embeddings from RS. A disadvantage is that it is less intuitive how one should train the new parameters together with the standard Transformer parameters. We have discussed this in Section 5; please check the paragraph titled “warm-start training for adapter-augmented LMs”.
> 8. P4LM can utilize available user or item information to generate relevant endorsement texts. For new items, as long as their descriptions are provided (as embeddings or as texts), P4LM can create appealing endorsements. For new users, as long as the behavioral embedding is given, P4LM can still generalize to produce personalized endorsements, otherwise the model focuses less on personalization.
> 9. Our work does not address the item recommendation aspect of conversational recommender systems. Instead, we focus on enhancing LLMs for generating endorsement texts of particular items. Our primary comparison is with PaLM2 baselines to assess improvements in Precision, Appeal, Personalization, and Preference Relevance.
> 10. Your question about the trade-offs between different reward models is pertinent. The full scope of these trade-offs can only be understood through extensive human evaluation studies. Future research could explore how to balance these weights to achieve desired endorsement outcomes with varying properties.
>
> We hope these responses satisfactorily address your concerns and questions. We are committed to making all necessary improvements based on your feedback.

---

> > ### Author Response · Authors · 2023-11-23
> > **Request to Reviewer wWN5**
> >
> > Dear Reviewer,
> >
> > Thank you so much for your time and efforts in reviewing our paper. We have addressed your comments in detail and are happy to discuss more if there are any additional concerns. We are looking forward to your feedback and would greatly appreciate you consider raising the scores. For an overview of the latest updates and changes we've made, please refer to our common response comment.
> >
> > Thank you,
> >
> > Authors

---

### Official Review · Reviewer_6DoX · 2023-11-04

**Soundness:** 3 good
**Presentation:** 3 good
**Contribution:** 2 fair
**Rating:** 5
**Confidence:** 4

**Summary:**

This article proposes that individuals should pay greater attention to the attributes of compelling, precision, personalization, and preference relevance when engaging in the recommendation process. Furthermore, the article introduces the P4LM model as a means to achieve this objective. To this end, the authors have meticulously designed reward functions for each attribute and utilized Reinforcement Learning with Adaptive Importance Sampling (RLAIF) to fine-tune the PALM model. Experimental evaluations were conducted on a subset of the MovieLens dataset and validated the ability of their method to improve the model performance.

**Strengths:**

1.	The proposed need for attention towards the compelling, precise, personalized, and preference-relevant directions for recommendation presented in this paper holds significant research significance.
2.	The paper provides a comprehensive description of the methodology, experimental details, and the utilization of prompts.

**Weaknesses:**

1.	Although the paper incorporates a plethora of metrics, the efficacy of these indicators in truly measuring the corresponding performance needs to be substantiated (see Questions for more details).
2.	The baseline used by the author is too concise. It is a consensus in the LLM field that models that have undergone reinforcement learning have better rewards than SFT and pre-trained models. More recommendation-related baselines should be mentioned.
3.	The article only collected data and trained on a closed-source model. The comparison of training on open-source models should be discussed.
4.	The author mentioned multiple ways to use PaLM to construct synthetic data in the article, but the rationality of this method has not been verified. (see Questions for more details)

**Questions:**

1.	According to Table 3, we can see that three of the four indicators proposed by the author (Precision, Personalization and Appeal) do not change significantly in different settings. On the other hand, compared with Table 2, the ordering between different settings cannot be maintained. Consistent, does this indicate that the metrics proposed by the author cannot measure the corresponding generated features? Besides, the Pref. Relevance of all models is very high. Does this mean that this indicator does not have discrimination?
2.	As I understand it, the author's model can generate recommended items and corresponding reasons. Can the author explain the advantages and disadvantages of recommended items compared to traditional recommendation models? I believe that comparable recommendation performance would be an acceptable outcome.
3.	The author mentions in the text the utilization of PaLM to construct data for training the reward models of Personalization and Preference Relevance. One question arises: despite the strong capabilities of LLM, there seems to be no conclusive evidence to suggest their proficiency in accomplishing this task effectively. Can the author provide corresponding evidence through real-world data or human evaluation?
4.	Due to the wealth of global knowledge and generalization capabilities possessed by LLM, could P4LM, after undergoing direct training in the movie domain, be applicable to other domains?

---

> ### Author Response · Authors · 2023-11-16
> **Author Response**
>
> __[Key Items]__
> * [Evaluation] Please see the common response regarding human evaluations and the experiments using the Amazon dataset.
> * [Comparison & Contributions] Please refer to 5 of the common responses.
>
> __[Our Detailed Responses]__
>
> Thank you for your review and the insightful questions raised. While we appreciate your concerns, we would like to clarify some misunderstandings about our work. Your input is crucial for us to refine our paper and accurately present our research.
>
> __[W1]__  We acknowledge that our evaluation can be further substantiated, and to address this, we are expanding our experiments to include the Amazon product dataset and conducting comprehensive human evaluations.
>
> __[W2]__ Our work uniquely addresses the generation of factual and personalized recommendation endorsement texts, which is distinct from the traditional item selection focus in recommendation literature. Our primary objective is novel, which is to enhance the quality of recommendation endorsements of specific items, not to recommend items to users. We are open to including comparisons with any relevant baselines that you suggest.
>
> __[W3]__ We agree that an extensive study involving various LLMs, both open-sourced and closed-source, would be valuable. However, our current focus is on demonstrating the effectiveness of our approach using the PaLM2 family of LLMs. Our aim is to establish the importance of our identified pillars for recommendation endorsements (such as Precision, Appeal, etc.) and to demonstrate improvements over baseline methods. We are supplementing our model-based evaluations with human evaluations and additional dataset analysis (currently in progress) to provide more comprehensive evidence of our approach's effectiveness.
>
> __[W4]__ The absence of datasets containing personalized recommendation endorsement texts presented a significant technical challenge. We leveraged recent advances in RLAIF to overcome this issue, using PaLM to generate synthetic data for training our reward models.
>
> __[Q1]__ Please refer to the last paragraph of Section 5 in our paper, where we discuss the differences in model-based and human evaluation results. We highlight that using multiple reward functions can mitigate issues of 'reward hacking'. The upcoming human evaluation results will further elucidate the relationship between model-based metrics and human perception of quality.
>
> __[Q2]__ We would like to reiterate that our work does not address the item selection problem in recommender systems. Our goal is to train an LLM to generate compelling, factual, personalized, and relevant endorsement texts for a given item and user.
>
> __[Q3]__ In response to your suggestion, we are conducting human evaluations and utilizing the Amazon product dataset to train P4LM.
>
> __[Q4]__ Your question about the cross-domain applicability of P4LM is intriguing. While there may be cases where preferences in one domain inform another, the ideal approach would be to train P4LM on a diverse set of domains to enhance its capability in generating personalized endorsement texts.
>
> We hope these responses clarify the aspects of our work and address your concerns. We are committed to enhancing the clarity and robustness of our research based on your feedback.

---

> > ### Author Response · Authors · 2023-11-23
> > **Request to Reviewer 6DoX**
> >
> > Dear Reviewer,
> >
> > Thank you so much for your time and efforts in reviewing our paper. We have addressed your comments in detail and are happy to discuss more if there are any additional concerns. We are looking forward to your feedback and would greatly appreciate you consider raising the scores. For an overview of the latest updates and changes we've made, please refer to our common response comment.
> >
> > Thank you,
> >
> > Authors

---

### Author Response · Authors · 2023-11-16
**Thank you for your reviews. Here's our common response (individual ones follow below).**

Thank you for your valuable feedback and insightful suggestions on our paper. We appreciate the opportunity to clarify and enhance our work during this author-reviewer discussion period. Here, we address the major concerns shared by you and provide updates on our ongoing efforts to strengthen the paper.

__[Major updates to the paper and evaluation]__
* __RM evaluation__: We used GPT-4 curated test sets for our RM evaluation. The __Appeal RM achieved 100% accuracy, and the Personalization RM reached 93.75% accuracy__, with each model tested against 32 examples. As soon as _human evaluation data is available_, we plan to use it to further validate the effectiveness of our RMs
* __Human evaluation__: We are currently conducting __extensive human evaluations__. In these, raters are asked to make __pairwise comparisons__ of endorsement texts generated by P4LM and baseline models, focusing on four specific criteria. We plan to update the paper with new results as soon as they become available.
* __Amazon dataset__: We are in the process of training models using the Amazon product dataset. Updates to the paper with fresh results and insights from these experiments will be provided as they become available.
* __Rewriting__: The paper has been revised for enhanced readability and clarity. Further revisions will be made to incorporate new results and discussions from the evaluations mentioned above.

__Below, we provide more detailed discussion.__

1. _Validation of reward models (RMs) using GPT-4 and subsequent human evaluations_:
In response to concerns about the alignment of our RMs with human preferences, we initially used GPT-4 to generate labels for test sets. Specifically, for the Appeal RM, we created a test set comprising 32 tuples of (movie description, worse endorsement text, better endorsement text). The RM’s effectiveness was gauged by its ability to distinguish the better endorsement text. Our evaluation yielded __an accuracy of 1.0 for the Appeal RM and 0.9375 for the Personalization RM__.

2. _Validation of RMs using human evaluations_:
To understand the characteristics of different RMs, we are conducting human rater evaluations specifically aimed at assessing the RMs. The raters will tell us which of the two endorsement texts given to them is better with respect to Precision, Appeal, Personalization, or Preference Relevance. This forms another test set with which we can evaluate the accuracies of the RMs. We will share the results of this evaluation as soon as we obtain them.

3. _Human rater evaluations for comparing P4LM and baselines_:
Beyond the existing human evaluations on different P4LM models, we are conducting human rater evaluations comparing P4LM, P4LM-S, SFT, SFT-Text, and PaLM2-L. These evaluations will provide a critical assessment of our model’s performance across various aspects, including Precision, Appeal, Personalization, and Preference Relevance. We will share these results and update the paper once they are available.

4. _Adding the Amazon product dataset for evaluation_:
We acknowledge the limitation of relying solely on the MovieLens dataset. To address this, we are now training our model (P4LM) on the public Amazon product dataset, which is significantly larger and more diverse, comprising 9.35M items with textual descriptions, 20.9M users, 233.1M reviews, and 82.83M ratings. This expansion will allow us to test our approach in a more varied and real-world context. We will update the paper with results from these tasks, including evaluations by human raters once they are complete.

5. _Clarification on the scope of our work_:
We would like to clarify an important misunderstanding. Our paper focuses on training an LLM to generate factual, compelling, personalized, and relevant recommendation endorsement  statements (or “pitches”) for a given item. Our aim is not to solve the general item recommendation problem. Thus, comparisons with existing baseline recommenders are not applicable to our setting.

6. _Paper revision for improved clarity_:
We have revised the paper to improve readability and clarity, with changes highlighted in red for easy identification.

7. _Individual responses to reviewer comments_:
We provide detailed responses for each specific question and comment from the reviewers individually below.

We believe these efforts and clarifications address the major concerns raised and significantly strengthen the contribution of our work. We look forward to further discussion and are committed to making all necessary updates in a timely manner.

---

> ### Author Response · Authors · 2023-11-23
> **New Updates**
>
> __[New Updates]__
>
> Dear Reviewers,
>
> We have updated our paper with additional human evaluation results and reward model evaluation outcomes. These updates are now included in the main text, with detailed information available in the revised sections of the appendix. We would like to highlight and re-emphasize some key results:
>
> * RM evaluation using endorsement texts generated by GPT-4 has yielded accuracies of  __1.0 for both the NLI and Appeal RMs, and 0.9375 for the Personalization RM.__
> * Human evaluation results:
>
> | Method | Precision | Personalization | Appeal | Preference Relevance |
> | --------- | --------- | --------- | --------- | --------- |
> SFT-Text | 0.29 | 0.458 | 0.375 | 0.458 |
> SFT | 0.458 | __0.583__ | 0.458 | 0.583 |
> P4LM | __0.5__ | __0.583__ | __0.667__ | __0.708__ |
> P4LM-S | __0.5__ | 0.5 | 0.5 | 0.333 |
>
> Here, we presented raters with two endorsement texts: one from PaLM2-L and another by a different model, to assess their relative performance. In Precision, P4LM and P4LM-S equaled PaLM2-L in raters' preferences, despite their smaller sizes (XS and XXS). Also, P4LM outperformed baselines in personalization, appeal, and preference relevance. These findings are based on 24 responses for each question.
>
> We kindly request the reviewers to re-evaluate our submission, taking into account these updates and the additional information provided during the author-reviewer discussion period. Your consideration and reassessment in light of these new findings would be greatly appreciated.
>
> Thank you for your continued engagement with our work!

---

### Meta-Review · Area_Chair_uGzL · 2023-12-06

**Metareview:**

The paper tackles the problem of applying reinforcement learning with AI feedback to fine-tune a pre-trained language model in the context of recommender systems. In this context, they consider four reward models that measure precision, appeal, personalization, and preference relevance of the generated responses. All the reviewers attributed some merit to the submission, however, they have a long list of concerns/points for improvement. While the authors did their best at addressing these concerns and implemented a lot of updates to their manuscript (e.g., adding a new set of experiments with human raters), the reviewers were not persuaded to change their overall assessment, even taking into account the clarified misunderstandings. As a result, I am unable to recommend acceptance.

**Justification For Why Not Higher Score:**

Significant number of concerns highlighted by the reviewers.

**Justification For Why Not Lower Score:**

N/A

---

### Decision · Program_Chairs · 2024-01-16

Reject